# Primary tumors release ITGBL1-rich extracellular vesicles to promote distal metastatic tumor growth through fibroblast-niche formation

Qing Ji[1,2,6], Lihong Zhou[1,2,6], Hua Sui[1,2,6], Liu Yang[1], Xinnan Wu[1,2], Qing Song[1,2], Ru Jia[1], Ruixiao Li[1,2], Jian Sun[1], Ziyuan Wang[1], Ningning Liu[1], Yuanyuan Feng[1], Xiaoting Sun[1,2], Gang Cai[1], Yu Feng[1], Jianfeng Cai[3], Yihai Cao[2,4], Guoxiang Cai[5✉], Yan Wang[1,2✉] & Qi Li[1,2✉]

Tumor metastasis is a hallmark of cancer. Metastatic cancer cells often reside in distal tissues and organs in their dormant state. Mechanisms underlying the pre-metastatic niche formation are poorly understood. Here we show that in a colorectal cancer (CRC) model, primary tumors release integrin beta-like 1 (ITGBL1)-rich extracellular vesicles (EVs) to the circulation to activate resident fibroblasts in remote organs. The activated fibroblasts induce the pre-metastatic niche formation and promote metastatic cancer growth by secreting pro-inflammatory cytokine, such as IL-6 and IL-8. Mechanistically, the primary CRC-derived ITGBL1-enriched EVs stimulate the TNFAIP3-mediated NF-κB signaling pathway to activate fibroblasts. Consequently, the activated fibroblasts produce high levels of pro-inflammatory cytokines to promote metastatic cancer growth. These findings uncover a tumor–stromal interaction in the metastatic tumor microenvironment and an intimate signaling communication between primary tumors and metastases through the ITGBL1-loaded EVs. Targeting the EVs-ITGBL1-CAFs-TNFAIP3-NF-κB signaling axis provides an attractive approach for treating metastatic diseases.

---

[1] Department of Medical Oncology and cancer institute, Shuguang Hospital, Shanghai University of Traditional Chinese Medicine, 201203 Shanghai, China. [2] Academy of Integrative Medicine, Shanghai University of Traditional Chinese Medicine, 201203 Shanghai, China. [3] Department of Chemistry, University of South Florida, Tampa, FL 33620, USA. [4] Department of Microbiology, Tumor and Cell Biology, Karolinska Institutet, Stockholm 171 77, Sweden. [5] Department of Colorectal Surgery, Fudan University Shanghai Cancer Center, 200032 Shanghai, China. [6]These authors contributed equally: Qing Ji, Lihong Zhou, Hua Sui. ✉email: gxcai@fudan.edu.cn; wangyan_sg@126.com; qili@shutcm.edu.cn

Colorectal cancer (CRC) is one of the most common malignancies and the 5-year survival rate of CRC patients with localized primary disease is ~90%. Approximately 70.4% of patients exists lymph node or peripheral metastasis, and 12.5% patients with distant metastasis[1]. CRC metastasis critically affects postsurgical survival rate and the mechanisms need to be elucidated[2].

Since proposing the "seed-and-soil" hypothesis[3,4], much progress has been made toward unearthing the mechanisms governing tumor metastasis. Recent emerging evidence shows the existence of complex communicating mechanisms between primary tumors and remote metastasis[5–7]. Inflammatory cells, immune cells, and hepatocytes participate in the formation of metastatic niche to support metastatic tumor growth[8–12]. Extracellular vesicles (EVs) are tumor-derived vehicles to mediate molecular communications between primary tumors and metastases[13,14]. EVs are small membrane vesicles, typically 30–100 nm in diameter, containing various functional biomolecules, such as proteins, lipids, RNAs, and DNAs, which could be horizontally transferred to recipient cells[15–17]. Integrins $\alpha6\beta4$ and $\alpha v\beta5$ in EVs are predictive markers for organ-specific metastasis[18]. In our present studies, we have found that integrin beta-like 1 (ITGBL1) was enriched in the plasma EVs of CRC patients with lung and liver metastasis.

ITGBL1 was first cloned and characterized from an osteoblast cDNA library. It encoded a ten integrin epidermal growth factor (EGF)-like repeat domain-containing protein[19]. It contains neither an RGD (Arg-Gly-Asp)-binding domain nor a transmembrane domain[20], suggesting that its function may be different from other integrins. ITGBL1 has been found to be overexpressed in metastatic bone cancer cells[21], and has been reported to mediate bone metastasis in breast cancer patients[22]. These findings suggest that ITGBL1 potentially contributes to developing cancer metastasis. Unlike integrins $\alpha6\beta4$ and $\alpha v\beta5$ in EVs, the role and molecular mechanism of ITGBL1-enriched EVs in CRC progression and metastasis remain unclear.

In this study, we uncover a molecular mechanism in which Runt-related transcription factor 2 (RUNX2) drives the transcription and secretion of ITGBL1-enriched EVs. The key function of cancer cell-derived ITGBL1-enriched EVs is involved in converting lung fibroblasts and hepatic stellate cells to an activated phenotype, such as myofibroblasts. Our data show a mechanism that underlies the primary CRC tumor-controlled metastatic growth in remote organs.

## Results

### ITGBL1-rich EVs in plasma correlate with CRC metastasis.
To study signaling molecules that might govern communication between primary tumors and metastasis, we performed an unbiased gene expression profiling analysis on CRC tissues (Fig. 1a). ITGBL1 was identified as one of the most overexpressed genes in primary CRC and metastatic sites relative to matched normal tissues (Fig. 1b). Real-time PCR analysis of 124 primary CRC, 46 metastatic tissue, and 124 adjacent normal tissue samples showed significantly higher ITGBL1 mRNA levels in 46 primary CRC tumors and matched metastatic sites than those (78 primary CRC tumors) in the nonmetastatic lesions, and far higher than those in all the adjacent normal tissues (Fig. 1c). The optimized cutoff value divided CRC patients into two groups: those with high and low ITGBL1 mRNA levels. The results demonstrated that CRC patients with lower ITGBL1 mRNA-expressing tumors had improved prognosis (Fig. 1d, e), and the mRNA levels of ITGBL1 were closely associated with CRC TNM (tumor, lymph node, and metastasis) stages, and liver and lung metastasis (Supplementary Table 1). Furthermore, the protein expression of

ITGBL1 was detected using tissue microarrays (TMAs), containing 124 CRC tissue and 124 adjacent normal tissue samples. The results in Fig. 1f showed that, in contrast to the adjacent tumor tissues, ITGBL1 was overexpressed in CRC tumor tissues, and its expression levels were reversely correlated with overall survival (OS) and disease-free survival (DFS) of CRC patients (Supplementary Fig. 1a, b).

ITGBL1 protein expression levels were further validated by immunohistochemical (IHC) staining. Again, significantly higher protein levels of ITGBL1 were found in primary CRC tumors, and matched lung and liver metastasis (Supplementary Fig. 1c, Supplementary Table 2). A The Cancer Genome Atlas (TCGA) dataset analysis also showed elevated ITGBL1 mRNA levels in 467 CRC primary tissues. The ITGBL1 mRNA expression was inversely correlated with OS (Supplementary Fig. 2a–c). A wide range of protein expression levels of ITGBL1 was also detected in 90 CRC tissues (only five cases overlapping with above 467 CRC cases), and the protein expressions of ITGBL1 were inversely correlated with lymphovascular invasion (Supplementary Fig. 2d).

Since ITGBL1 protein is highly homologous to the N-terminal EGF-like stalk fragment of integrin $\beta$ (ref. [20]), we questioned whether ITGBL1 protein is released into the EVs like integrins $\alpha6\beta4$ and $\alpha v\beta5$ to promote distant metastases. For this purpose, EVs from the plasma of 124 CRC patients and 32 healthy volunteers (normals) were purified by sequential centrifugation as previously described (Fig. 1g–i)[14]. An enzyme-linked immunosorbent assay (ELISA) assay showed that ITGBL1 was significantly higher in the plasma EVs of CRC patients (Fig. 1j), with particularly higher values in patients with distant metastases (Fig. 1k). Moreover, the mRNA expression of ITGBL1 in human CRC tumor samples had strong correlation with the ITGBL1 levels in corresponding plasma EVs of CRC patients (Supplementary Fig. 3a). The ITGBL1 levels in EVs were inversely correlated with OS and DFS (Fig. 1l, m). Multivariate Cox regression analysis further revealed that both ITGBL1 levels in EVs and distant metastasis served as independent predictors of poor prognosis in CRC patients (Supplementary Table 3). In addition to ITGBL1, several other integrins, including ITGα6, ITGβ4, and ITGβ1 were also packed in the plasma EVs of CRC patients with lung metastases, whereas ITGα5 and ITGβ5 were found in liver metastases (Supplementary Fig. 3b–f). These findings support that integrins $\alpha6\beta4$ and $\alpha v\beta5$ in EVs are found in organ-specific metastasis[18].

### RUNX2-dependent regulation and secretion of ITGBL1-rich EVs.
According to the TCGA dataset (Fig. 2a, left panel), we found that ITGBL1 was co-expressed with a set of genes related to metastasis and cell adhesion, such as transcription factor RUNX2, integrin ITGB1, adhesion molecules FN1 (fibronectin 1), COL8A2 (collagen, type VIII, alpha 2), MMP9 (matrix metalloprotein 9), chemokine CXCL12 (chemokine (C-X-C motif) ligand 12), and others. In the right panel of Fig. 2a, highly matched co-expression of ITGBL1 and RUNX2 mRNA was found both in 467 CRC cases from TCGA dataset and our 124 CRC cases. RUNX2 is a member of the RUNX family of transcription factors and encodes a nuclear protein with a Runt DNA-binding domain[21]. Its function in CRC remains unknown. In our previous work, we have investigated the expression levels of RUNX2 in 124 CRC tumor specimens, and the results demonstrated that RUNX2 mRNA levels were significantly higher in primary tumors and in metastatic sites than in nonmetastatic tumors, and its mRNA levels were inversely correlated with CRC TNM stages, and liver and lung metastasis[23]. Survival analysis also concluded that patients with lower RUNX2 mRNA expression had prolonged OS and

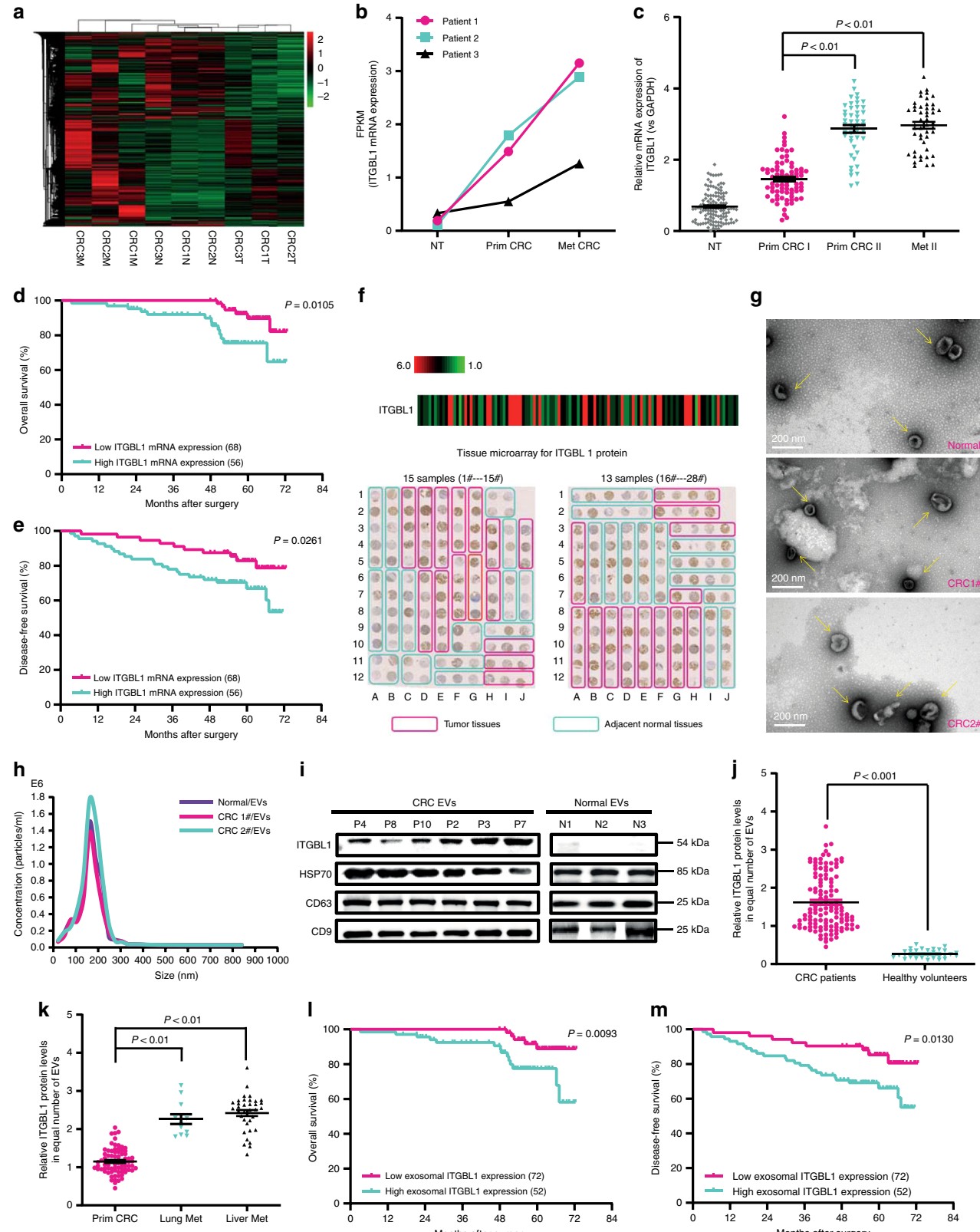

DFS[23]. Since *ITGBL1* was co-expressed with *RUNX2*, a potentially important transcription factor for ITGBL1 (ref. [24]), we next investigated the regulatory mechanism by which RUNX2 induced ITGBL1 expression. We constructed a dual-luciferase reporter vector containing the *ITGBL1* promoter region between −825 and +36, the regulatory transcription elements for RUNX2

(Fig. 2b). Dual-luciferase reporter assays demonstrated that RUNX2 enhanced the *ITGBL1* promoter activity, which was decreased with knockdown (KD) of RUNX2. Mutation of the *ITGBL1* promoter abolished the regulatory activity of RUNX2 (Fig. 2c). A chromatin immunoprecipitation (ChIP) assay showed that RUNX2 bound to the *ITGBL1* promoter region containing

**Fig. 1 ITGBL1 expression in postsurgical, primary, and metastatic sites, compared with primary sites of nonmetastatic CRC patients. a** RNA sequencing and cluster analysis of differentially expressed mRNAs in three paired tissues (primary CRC1T, CRC2T, CRC3T, and metastatic CRC1M, CRC2M, CRC3M) of metastatic CRC patients and matched normal tissues (CRC1N, CRC2N, and CRC3N). Red represents high expression and green represents low expression. **b** Quantitative analysis of *ITGBL*1 mRNA according to the FPKM (fragments per kilobase of exon per million fragments mapped) data from RNA sequencing. **c** Expression levels of *ITGBL1* mRNA in 124 CRC primary tumor, 46 metastatic, and 124 adjacent normal tissues were analyzed by qRT-PCR. NT represents adjacent normal tissues. **d, e** Kaplan–Meier analyses of the correlations between *ITGBL1* mRNA expression and OS and DFS. **f** Tissue microarray results from 28 of 124 primary CRC patients' tumor tissues and adjacent normal tissues. 2–5 (average, 4) random, representative 0.6-mm cores were obtained from each case. Magenta frame: tumor tissues, turquoise frame: adjacent normal tissues. Left panel: 15 samples (1#–15#), right panel: 13 samples (16#–28#). **g, h** EVs from CRC patients' and healthy volunteer's plasma were analyzed by electron microscopy and LM10 nanoparticle characterization system. Yellow arrows indicate representative EVs. Scale bar, 200 nm. **i** Western blot analysis of ITGBL1 protein and EVs markers HSP70, CD63, and CD9 in plasma EVs from CRC patients and healthy volunteers (normal). P4, P8, and P10: three nonmetastatic CRC patients; P2, P3, and P7: three metastatic CRC patients; and N1, N2, and N3: three healthy volunteers (normal). **j** Relative ITGBL1 levels (ng of ITGBL1 per μg of EVs) in the plasma EVs of above 124 CRC patients and 32 healthy volunteers. **k** Relative ITGBL1 levels (ng of ITGBL1 per μg of EVs) in the plasma EVs of above 124 CRC patients. Primary CRC without metastasis: 78 cases; Lung Met: 11 cases; Liver Met: 35 cases. **l, m** Kaplan–Meier analyses of the correlations between ITGBL1 EVs expression and OS or DFS. Each experiment was performed three times independently and the results are presented as mean ± SD. Student's *t*-test was used to analyze the data; *p < 0.05, **p < 0.01.

the −825/+36 site, but not the other sites (Fig. 2d). Genetic gain- and loss-of-function experiments further supported the finding that RUNX2 may regulate *ITGBL1* mRNA and protein expression (Supplementary Fig. 4a, Fig. 2e).

We determined whether ITGBL1 level in EVs was also affected by RUNX2. ITGBL1 was enriched in the EVs of RUNX2-overexpressing cells, and KD of RUNX2 in cells ablated ITGBL1 levels in EVs (Fig. 2f, g). Conversely, in RUNX2 KD cells, the ITGBL1 levels in EVs was not altered when ITGBL1 was overexpressed or KD (Fig. 2h–j). The total numbers of EVs from an equal number of cells were also upregulated by overexpressing RUNX2, whereas KD of RUNX2 produced the opposite effect (Supplementary Fig. 4b, Fig. 2k). RUNX2 regulated the expression level of neutral sphingomyelinase 2 (nSMase2) encoding gene *Smpd3*. nSMase2 is known to regulate the biosynthesis of ceramide and trigger the secretion of small membrane vesicles[25]. To determine the transcription regulation of the *Smpd3* promoter by RUNX2, a 1.1 kb region flanking the 5′-UTR of *Smpd3* was cloned. A luciferase vector containing the 1000 nt to the +100 nt of the *Smpd3* promoter region or the *Smpd3* promoter-less region was constructed (Fig. 2l). The luciferase activity of *Smpd3* 5′-UTR was upregulated by RUNX2, and KD of RUNX2 ablated the activity (Fig. 2m). The mRNA and protein levels of nSMase2 were also found to be upregulated in RUNX2-overexpressing cells and downregulated in RUNX2 KD cells (Fig. 2n). Additionally, the total numbers of EVs from equal number of cells were also upregulated or downregulated by nSMase2 overexpression or KD (Fig. 2o). ITGBL1 levels in EVs from an equal number of cells were upregulated in nSMase2-overexpressing cells, but down-regulated in nSMase2 KD cells (Supplementary Fig. 4c, Fig. 2p). Treatment with GW4869, a known inhibitor for ceramide biosynthesis[26], resulted in a dose-dependent reduction of ITGBL1 levels in EVs from equal number of cells (Fig. 2q). These results suggested that a ceramide-dependent secretion pathway was involved in the RUNX2-mediated secretion of ITGBL1-loaded EVs.

**CRC-ITGBL1-enriched EVs promote fibroblast activation.** In the pre-metastatic niche or the metastatic niche, cancer-associated fibroblasts (CAFs) have been demonstrated to involve in the progression of tumor metastasis or metastatic growth[27]. Considering that EVs might affect the tumor micro-environment (TME), we investigated the role of ITGBL1-loaded EVs in activating CAFs (ref. [28]).

To study the tissue distribution of EVs, purified EVs were intravenously injected into mice according to the protocols provided by Hoshino et al.[18]. In vivo tracing of the injected EVs

showed that CRC-derived EVs were accumulated in lung or liver, and very weak signals in the brain and bone marrow (Fig. 3a, Supplementary Fig. 5a). Importantly, immunofluorescence ana-lysis showed that CRC-derived EVs were efficiently taken up by α-smooth muscle actin + hepatic stellate cells (α-SMA + hStCs), myofibroblasts, S100A4 + fibroblasts, and F4/80+ macrophage cells, but not by CD31 + endothelial cells in the liver and lung (Fig. 3b). Control EVs isolated from normal colonic NCM-460 epithelial cells was not efficiently taken up by these cell types (Fig. 3b). These data suggested that α-SMA + hStCs, myofibro-blasts, S100A4 + fibroblasts, and F4/80+ macrophage cells were involved in the initial steps of pre-metastatic niche formation in the liver and lung. CAFs are known to secrete the pro-inflammatory cytokines, including interleukin 6 (IL-6), IL-8, and IL-1β, and express transforming growth factor β (TGF-β), α-SMA (a typical myofibroblastic marker), and chemokines, such as CXCL12 (refs. [27–29]). We isolated the primary fibroblasts and stellate cells from mice lung and liver, receiving treatment of tumor-derived EVs. IL-6, IL-8, and IL-1β pro-inflammatory cytokines, and fibrotic markers, such as TGF-β, α-SMA, and CXCL12 were all upregulated in ITGBL1-enriched EVs-treated fibroblasts and stellate cells relative to control NCM-460-EVs-treated cells (Fig. 3c, d). Notably, CRC-ITGBL1-enriched EVs increased the frequency of α-SMA + hStCs, myofibroblasts, FN deposition, and recruitments of F4/80+ macrophage, and Ly6G+ myeloid cells to the liver (Fig. 3e, Supplementary Fig. 5b–e) or lung (Fig. 3f, Supplementary Fig. 5f–i). These results demon-strated that ITGBL1-enriched EVs from CRC cells established a pre-metastatic niche formation in the liver and lung.

Mice were pretreated with EVs containing different levels of ITGBL1 for 3 weeks, following by intravenous injection of tumor cells as experimental metastasis models. In vivo imaging analysis showed that purified ITGBL1-enriched EVs promoted the growth of lung metastatic tumors, whereas silencing ITGBL1 markedly decreased the metastatic growth (Fig. 4a). Histological analysis further validated the metastatic promoting effect by ITGBL1-enriched EVs and silencing ITGBL1 decreased the lung metastatic foci (Fig. 4b). Similarly, the experimental liver metastasis model also demonstrated the promoting effect of metastatic cancer by ITGBL1-enriched EVs (Fig. 4c, d). In an orthotopic CRC model by which cancer cells were implanted into the cucum, ITGBL1-enriched EVs also accelerated liver metastatic tumor growth (Fig. 4e, f). Nearly all the CRC-EVs-treated mice developed liver metastases, whereas only 16.7–33.3% of the CRC-EVs-treated mice developed lung metastases. In contrast to liver and lung, the brain and bone tissues lacked detectable metastases (Supplementary Fig. 6a, b).

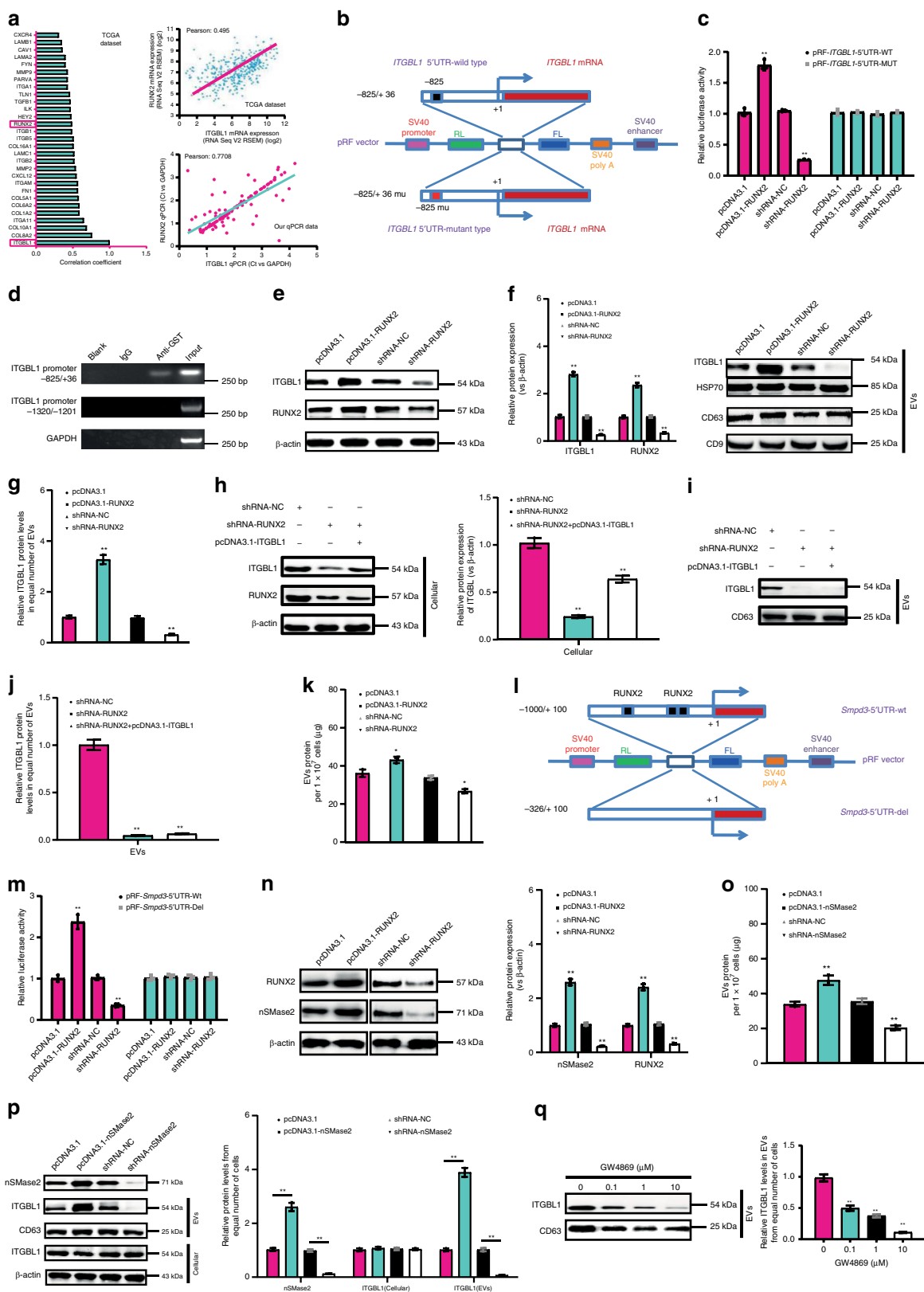

Immunofluenrescence assay of α-SMA, a known myofibroblastic marker, demonstrated that treatment with ITGBL1-enriched EVs increased the α-SMA + CAFs in metastatic lesions (Fig. 4g, Supplementary Fig. 6c–e). CAFs isolated from above lung and liver metastatic lessions expressed high mRNA levels of *IL-6*, *IL-8*, *IL-1β*, *α-SMA*, *TGF-β*, and *CXCL12* (Fig. 4h–j). These results

preliminary demonstrated that ITGBL1-enriched EVs derived from highly metastatic cancer cells accelerate metastatic cancer growth through a fibroblasts activation mechanism.

**ITGBL1-enriched EVs promote fibroblasts activation in vitro.**
Knowing that EVs secreted from cancer cells is an important

**Fig. 2 RUNX2 drove the transcription and secretion of ITGBL1-loaded EVs in CRC. a** Genes co-expressed with *ITGBL1* based the RNA sequencing data in TCGA dataset (left panel). *ITGBL1* and *RUNX2* were both marked in the red frame. Right panel showed the co-expression of *ITGBL1* and *RUNX2* mRNA in 467 cases from TCGA RNA sequencing dataset, and 124 cases in our real-time PCR data. Pearson correlation coefficient was used to evaluate the correlation. **b, c** *ITGBL1* promoter activities with or without wild-type RUNX2-binding motif or the mutant RUNX2-binding motif were evaluated using a dicistronic vector pRF and dual-luciferase reporter assay. **d** Recruitment of RUNX2 to the promoter of *ITGBL1* in RUNX2-GST-overexpressing SW620 cells were detected by ChIP. GST antibody, control IgG, and special primers for *ITGBL1* promoter were appliedantibody. **e–g** ITGBL1 cellular and EVs expression assay in RUNX2-overexpressing cells and RUNX2-silent cells compared with their respective control cells. HSP70, CD63, and CD9 were protein markers for EVs. **h–j** The regulatory effect of ITGBL1 overexpression on the ITGBL1 cellular and EVs expression in RUNX2-knockdown SW620 cells. **k** Total proteins analysis in EVs from equal numbers of RUNX2-overexpressing cells or RUNX2-silent SW620 cells relative to their respective control cells. **l** The 1.1 kb ′-flanking promoter region activities of *smpd3* with or without the wild-type RUNX2-binding motif, or the deleted RUNX2-binding motif evaluated using a dicistronic vector pRF and dual-luciferase reporter assay. **m** *Smpd3* promoter activities in RUNX2-overexpressing cells and RUNX2-silent cells were compared with their control SW620 cells. **n** The effect of RUNX2 on the protein expression of nSMase2 (*smpd3* encoding protein) in SW620 cells. **o, p** Total proteins assay and special ITGBL1 expression assay in EVs from equal numbers of nSMase2-overexpressing cells or nSMase2-silent SW620 cells in compared with their respective control cells. **q** Effect of GW4869 (an nSMase inhibitor) on the secretion of ITGBL1-loaded EVs in equal numbers of SW620 cells treated with different concentrations of 0.1 μM, 1 μM, and 10 μM. Each experiment was performed three times independently and the results are presented as mean ± SD. Student's *t*-test was used to analyze the data; *$p < 0.05$, **$p < 0.01$.

communicator between primary tumors and distant metastasis, and crucial roles of CAFs in tumor metastasis and growth, we asked the question of whether ITGBL1-enriched EVs participate in the activation of fibroblasts. Among various CRC cell lines, ITGBL1 was more highly expressed in the EVs of SW620 and LoVo, relative to others (Supplementary Fig. 7a). The structure, size, and numbers of the isolated EVs were identified by electron microscopy and Nanosight particle tracking analysis (Supplementary Fig. 7b, c). Especially, abundant EVs were secreted from high-metastatic cancer cells than low-metastatic ones through a quantitative analysis of EVs isolated from an equal number of cells (Supplementary Fig. 7c). HSP70, CD63, and CD9, characteristic proteins for EVs further validated the isolated EVs (Supplementary Fig. 7d, e). EVs isolated from normal colonic epithelial cells NCM-460 were chosen as the control EVs and their structures, sizes, numbers, and characteristics were shown in Supplementary Fig. 7f–h.

Human WI-38 lung fibroblasts and LX-2 hepatic stellate cells were used in our models. To monitor the delivery of EVs, we labeled the CRC cells-derived EVs with PKH67 (green) and fibroblasts with PKH26 (red), respectively. After incubation, confocal imaging showed the presence of PKH67 spots in recipient fibroblasts and stellate cells, suggesting that EVs released by different tumor cells could be successfully delivered to fibroblasts and stellate cells in vitro (Fig. 5a, b, Supplementary Fig. 8a, b). Next, we paired the four CRC cell lines into the following groups: LoVo versus Caco2 (different origins) and SW620 versus SW480 (the same origin) to further evaluate their abilities in educating fibroblasts or stellate cells. Consistent with in vivo data (Fig. 4h–j), the highly metastatic CRC cell-EVs-educated fibroblasts and stellate cells upregulated pro-inflammatory genes, myofibroblastic markers, and chemokines, including *IL-6*, *IL-8*, *IL-1β*, *α-SMA*, *TGF-β*, and *CXCL12* (Supplementary Fig. 9a–d). More important, elevated mRNA levels of *IL-6*, *IL-8*, *IL-1β*, *α-SMA*, *TGF-β*, and *CXCL12* were detected in SW480- or SW620-derived high ITGBL1-EVs-treated fibroblasts and stellate cells, but ITGBL1-silencing ablated these effects (Supplementary Fig. 10a–d). The above results showed that cancer cell-derived ITGBL1-enriched EVs promoted the activation of fibroblasts and stellate cells in vitro.

**ITGBL1-enriched EVs activate fibroblasts via NF-κB signaling.** To further investigate a causative link between ITGBL1-enriched EVs and inflammation, we firstly investigated the nuclear factor kappa-B (NF-κB) signaling pathways. As shown in the Supplementary Fig. 11a, b, highly metastatic CRC-derived EVs increased the expression of phosphorylated NF-κB, decreased IkappaBalpha

(IκBα), and activated NF-κB signaling in WI-38 and LX-2. Detection of the relative luciferase activity of NF-κB in WI-38 and LX-2 further validated these findings (Supplementary Fig. 11c, d).

We next studied the roles of ITGBL1-loaded EVs in pro-inflammatory response. In Fig. 5c–f, ITGBL1-enriched EVs elevated the expression levels of phosphorylated NF-κB, inhibited IκBα, and activated NF-κB signaling, whereas ITGBL1 silencing produced opposing effects. Luciferase activity of NF-κB further validated these findings (Fig. 5g, h). Overexpression of ITGBL1 in WI-38 or LX-2 cells promoted NF-κB phosphorylation and IκBα depression (Fig. 5i). The relative luciferase activity of NF-κB also showed the positive regulatory effect of ITGBL1 (Fig. 5j). These data indicate that ITGBL1 promotes fibroblasts activation through a mechanism of NF-κB signaling activation.

**Activation of fibroblasts by ITGBL1-TNFAIP3-NF-κB signaling.** To study the mechanism underlying ITGBL1-induced NF-κB activation, we performed co-immunoprecipitation (Co-IP) in combination with liquid chromatography–tandem mass spectrometry (LC-MS/MS) experiments to identify potential binding partners of ITGBL1 protein (Fig. 6a). LC-MS/MS analysis identified several potential binding partners of ITGBL1 protein (Fig. 6b). Co-IP and immunoblot experiments discovered tumor necrosis factor (TNF) alpha-induced protein 3 (TNFAIP3) as a strong binding partner of ITGBL1 in WI-38 or LX-2 cells (Fig. 6c, d). TNFAIP3 is known to be rapidly induced by the TNF, and its encoded protein is a zinc finger protein and ubiquitin-editing enzyme[30]. Consistent with the Co-IP results, the immuno-fluenrescence analysis also showed co-location of ITGBL1 and TNFAIP3 (Fig. 6e, f). Treatment of fibroblasts or stellate cells with ITGBL1-enriched EVs from CRC cells showed co-location of ITGBL1 with TNFAIP3 by Co-IP and immunofluorescence detection (Supplementary Fig. 12a–d). Additionally, some other integrins have been reported to play important roles in promoting tumor metastasis[18], igniting our interest to observe whether the function of ITGBL1 is affected by other integrins. However, our Co-IP in combination with LC-MS/MS experiments have shown no direct and significant binding between ITGBL1 and other integrins. The subsequent Co-IP in combination with western blot experiments further validated no direct binding between ITGBL1 and other integrins (Supplementary Fig. 13a, b). Moreover, KD of integrin β1 barely affected the regulatory effect of ITGBL1 on the NF-κB signaling pathway (Supplementary Fig. 13c, d).

Subsequently, we investigated the possibility of TNFAIP3 in affecting ITGBL1-triggered activation of the NF-κB signaling pathway. Figure 6g–j showed that silencing of TNFAIP3

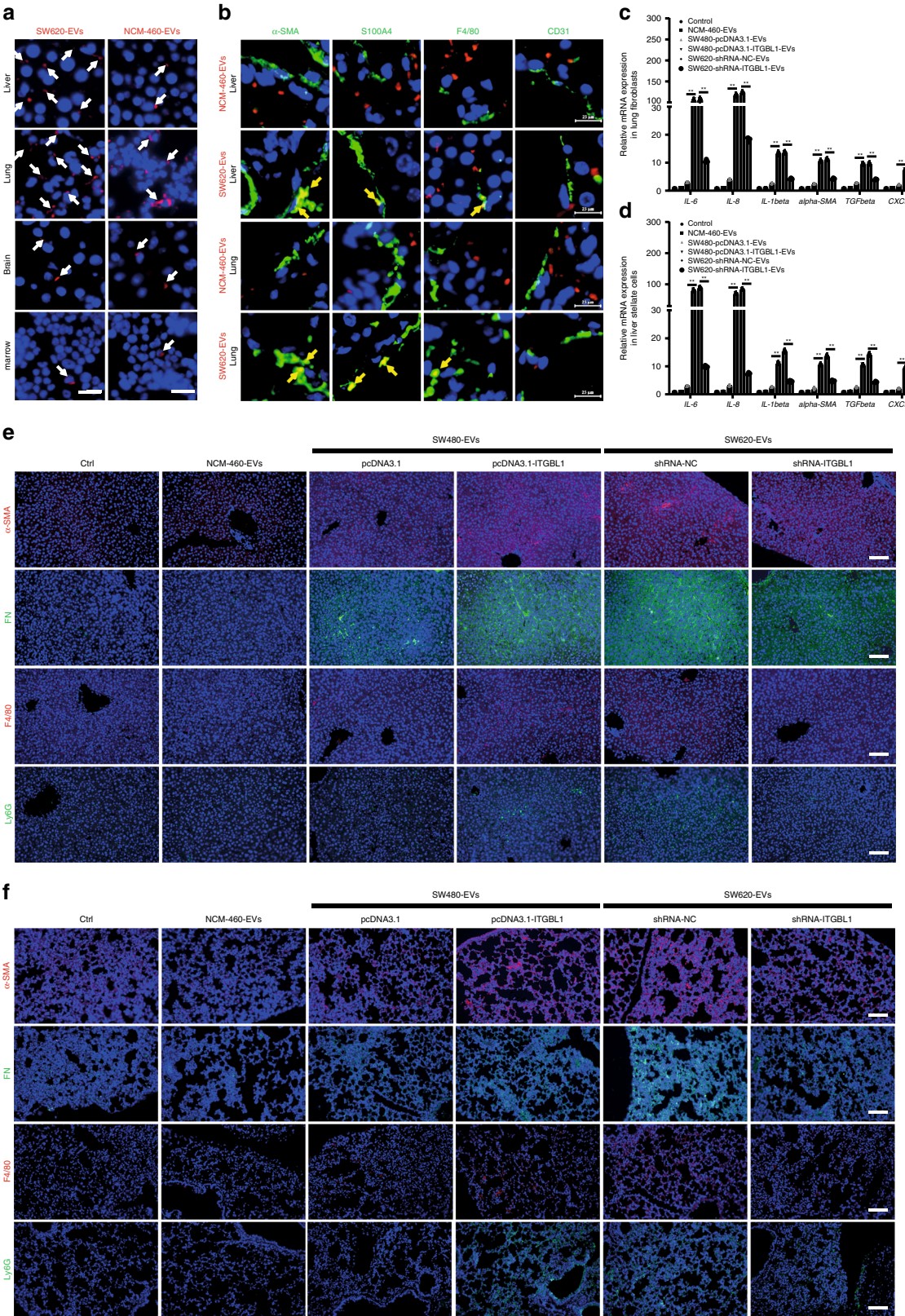

decreased ITGBL1-induced phosphorylation of inhibitor of nuclear factor kappa-B kinase alpha/beta (IKKα/β), and phosphorylated NF-κB, but increased IκBα. The relative luciferase activity of NF-κB also confirmed the regulatory effect of TNFAIP3 on the NF-κB signaling pathway (Fig. 6k). These results demonstrated that the fibroblasts

become activated via the ITGBL1-TNFAIP3-NF-κB signaling pathway.

**Activated fibroblasts promoted CRC progression.** Next, we investigated the impact of ITGBL1-overexpressing fibroblasts on

**Fig. 3 EVs derived from ITGBL1-overexpressing CRC cells induced the liver and lung pre-metastatic niche formation. a** Biodistribution of cancer-derived EVs in the potential metastatic organs of naive mice. Fluorescence microscopy analysis of the PKH26-labeled EVs (red) in the liver, lung, brain, and bone marrow of mice injected with SW620-EVs. Normal colonic epithelial cells NCM-460-EVs were used as the controls. White arrows indicate EVs foci. For each group, three mice were used for analysis. Scale bar, 50 μm. **b** Interaction between cancer-derived EVs and resident cells. NCM-460-EVs were used as the controls. Immunofluorescence analysis of resident tissue-specific stomal cells in the lung and liver of mice after injection of PKH26-labeled EVs (red). The 6 μm O.C.T. (Optimum Cutting Temperature) tissue cryosections were stained with antibodies against α-SMA, S100A4, F4/80, and CD31. Secondary antibodies conjugated to Alexa Fluor 488 (green) were used for imaging. Nuclear staining was done with DAPI (40, 6-diamidino-2-phenylindole). Top two: representative images of immunofluorescence microscopy of NCM-460-EVs or SW620-EVs co-staining with α-SMA, S100A4, F4/80, and CD31 in liver tissue sections from mice. Bottom two: representative images of immunofluorescence microscopy of NCM-460-EVs or SW620-EVs co-staining with α-SMA, S100A4, F4/80, and CD31 in lung tissue sections from mice. For each group, three mice were used for analysis. Scale bar, 25 μm. **c, d** Indicated genes (*IL-6*, *IL-8*, *IL-1β*, *α-SMA*, *TGF-β*, and *CXCL12*) expressions in the lung fibroblasts and liver stellate cells of mice in **b** were detected by qRT-PCR analysis. **e, f** Representative immunofluorescence images of α-SMA and FN expression in arbitrary units (a.u.), F4/80$^+$ cells, and Ly6G$^+$ cells in the liver and lung of the mice educated with no EVs (control), NCM-460-EVs, SW480-pcDNA3.1-EVs, SW480-pcDNA3.1-ITGBL1-EVs, SW620-shRNA-NC-EVs, and SW620-shRNA-ITGBL1-EVs. For α-SMA and F4/80 detection, secondary antibodies conjugated to Alexa Fluor 594 (red) were used. For FN and Ly6G detection, secondary antibodies conjugated to Alexa Fluor 488 (green) were used. Nuclear staining was done with DAPI. Scale bar, 100 μm. Each experiment was performed at least in triplicate. Student's *t*-test was used to analyze the data; *$p < 0.05$, **$p < 0.01$.

CRC progression. ITGBL1 overexpression enhanced the secretion of IL-6 and IL-8 in fibroblasts or stellate cells (Fig. 7a, b). Treatment of HCT116 cells with conditioned medium derived from ITGBL1-overexpressing WI-38 or LX-2 enhanced stemness genes expression and the spheroid formation ability (Fig. 7c–f). Blocking IL-6 or IL-8 with a neutralizing antibody partially reversed the increased spheroid formation ability of HCT116 cells (Fig. 7g, h). Furthermore, using the conditioned medium-pre-educated HCT116 cells, we found that the same conditioned medium in vivo promoted the tumorigenicity of CRC (Fig. 7i, j). In the subcutaneous transplantation tumors, we further confirmed the enhancement of stemness genes expression in tumors treated with conditioned medium from ITGBL1-overexpressing WI-38 or LX-2 (Supplementary Fig. 14a).

The expression of epithelial–mesenchymal transition (EMT)-associated genes was also altered in the conditioned medium-treated HCT116 cells, leading to the downregulation of E-cadherin and upregulation of Vimentin, Snail, and FN (Fig. 8a, b). As shown in Fig. 8c, d, the conditioned medium-treated HCT116 cells showed increased motility, but blocking IL-6 or IL-8 with their respective neutralizing antibodies partially reversed this increased motility (Supplementary Fig. 15a, b). In vivo, using the conditioned medium-pre-educated HCT116 cells, we built the experimental lung and liver metastasis models by tail vein injection or intra-splenic injection, and found that the same conditioned medium promoted CRC metastasis and growth in the lung and liver (Fig. 8e, f). Orthotopic implantation of the conditioned medium-pre-educated HCT116 cells in the cecum validated the stimulatory effect of metastatic growth in the liver (Fig. 8g). Histological analysis demonstrated the promoting effect of conditioned medium from ITGBL1-overexpressing WI-38 or LX-2 on the growth of liver metastatic tumors (Fig. 8h). In the lung and liver metastatic tumors from Fig. 8e, f, we further confirmed the positive adjustment of EMT-associated genes expression in metastatic tumors pretreated with conditioned medium from ITGBL1-overexpressing WI-38 or LX-2 (Supplementary Fig. 14b, c). These results demonstrated that ITGBL1-overexpressing fibroblasts activated and promoted CRC progression by secreting pro-inflammatory cytokines IL-6 and IL-8, and promoting stemness and EMT of CRC cells.

**Primary CAFs contribute to CRC progression.** Using above lung and liver metastatic tissues from CRC patients, we detected the α-SMA (characteric marker for myofibroblastic CAFs) expression by IHC staining analysis, and the results showed high density of α-SMA + myofibroblasts around the metastatic tumors

(Supplementary Fig. 16), implying the important function of α-SMA + myofibroblasts on metastatic CRC development. To further study the function of CAFs in CRC progression, we isolated primary CAFs from the liver metastaic tumors of CRC patients and primary normal fibroblasts (NFs) from healthy liver tissues. We analyzed the expression levels of ITGBL1 in the primary NFs and CAFs from several different CRC patients. CAFs with low ITGBL1 and high ITGBL1 expression were applied for further experimentation (Supplementary Fig. 17a, b). Expectedly, primary CAFs with high ITGBL1 expression secreted higher levels of IL-6 and IL-8 relative to low ITGBL1-expressing CAFs and NFs (Supplementary Fig. 17c, d). Similar to fibroblasts educated by ITGBL1-enriched EVs, primary CAFs with high ITGBL1 expression promoted tumorigenicity (Supplementary Fig. 17e, f), tumor stemness (Supplementary Fig. 17g), metastatic growth (Supplementary Fig. 17h, i), and EMT (Supplementary Fig. 17j). In vitro experiments validated the fact that primary CAFs with high ITGBL1 increased expression levels of stemness-related genes and the spheroid formation (Supplementary Fig. 18a, b). Additionally, ITGBL1 increased the expression levels of EMT-associated genes and tumor cells migration (Supplementary Fig. 18c, d). All above findings suggested that the primary CAFs with high ITGBL1 expression contribute to CRC progression through the similar mechanism as above ITGBL1-overexpressing fibroblasts did.

In summary, similar mechanism underlying interactions between primary CRC cells and distal fibroblasts through the EVs-mediated transfer of ITGBL1 may exist in promoting the metastasic niche formation in lung and liver. We provide compelling evidence to establish a signaling pathway by which tumor-derived ITGBL1-enriched EVs convert primary fibroblasts to CAFs via binding to TNFAIP3 and activating the NF-κB signaling pathway. The ITGBL1-activated CAFs promote CRC stemness, aggressiveness, and EMT by pro-inflammatory cytokines, such as IL-6 and IL-8. CAFs manipulated the TME to support metastatic tumor growth (Fig. 9).

## Discussion

Most studies have focused on understanding the mechanism underlying cancer metastasis[31–36]. Our present study provides clinical and experimental evidence in support of ITGBL1-enriched EVs-mediated CRC metastatic growth. As an extra-cellular matrix protein, ITGBL1 was first cloned from HUVEC (human umbilical vein endothelial cells), fetal lung, and osteoblast cDNA libraries[19]. ITGBL1 has been reported to promote breast cancer bone metastasis by activating the TGF-β signaling pathway[24]. In ovarian cancer, ITGBL1 promotes migration and

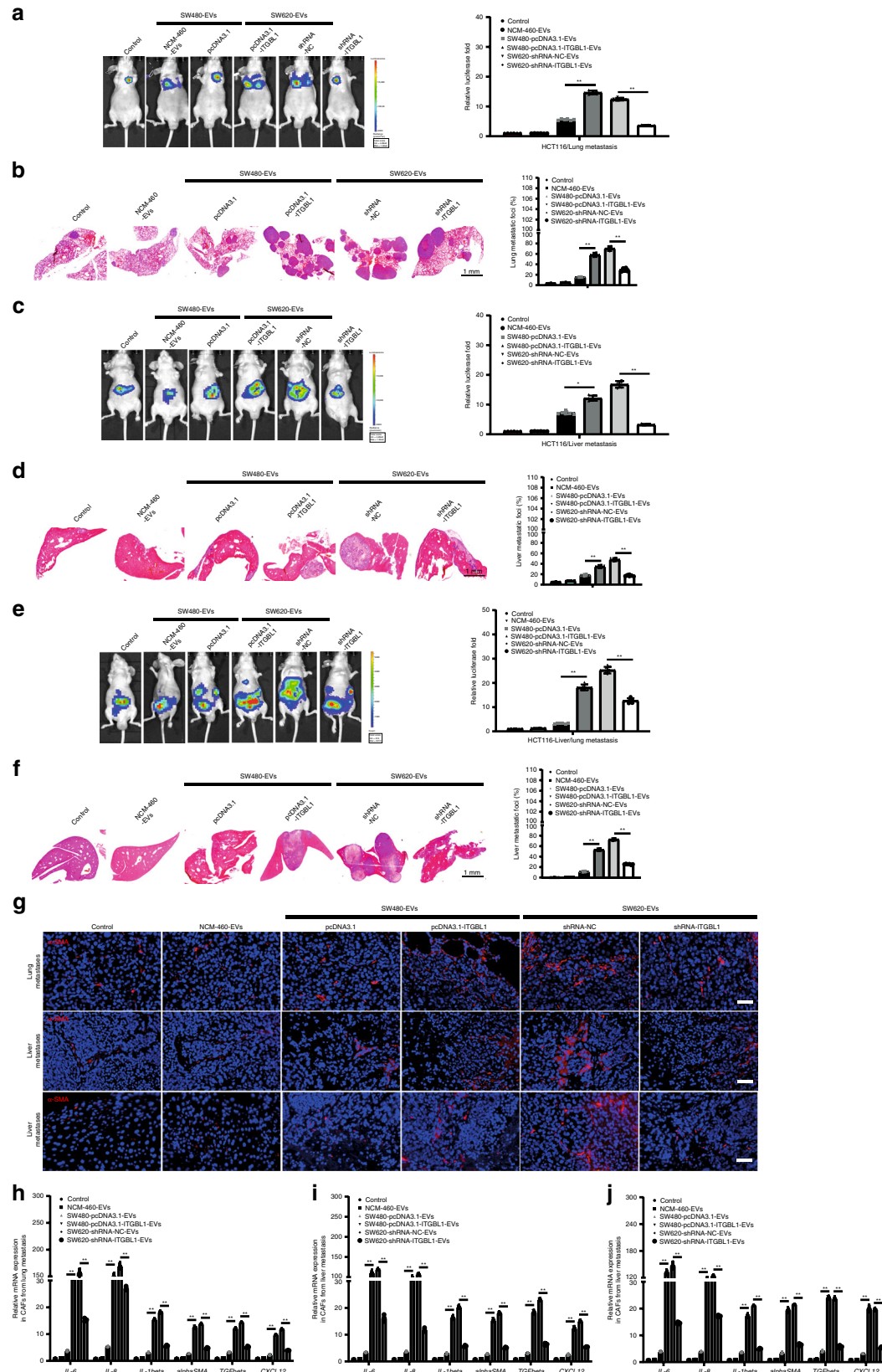

adhesion through Wnt/PCP (planar cell polarity) and FAK/SRC (focal adhesion kinase/steroid receptor coactivator) signaling pathways[37]. Here, our data demonstrates that ITGBL1 is highly expressed in advanced primary CRC tissues and is correlated with a metastatic phenotype.

Tumor-derived EVs often prepare favorable microenvironment for tumor metastasis or metastatic growth and are recognized as a critical determinant of tumor progression. Intriguingly, EVs are vehicles to mediate communications by carrying intercellular exchanges of proteins, mRNAs, and non-coding RNAs[15–17]. EVs

**Fig. 4 EVs derived from ITGBL1-overexpressing CRC cells increased the liver and lung metastasis. a, c** Luciferase-based bioluminescence imaging on experimental lung and liver metastasis of indicated mice treated with EVs derived from ITGBL1-overexpressing or -silent CRC cells relative to their respective control cells, or blank controls or normal colonic epithelial cells NCM-460. Luciferase-labeled HCT116 cells were used to perform experimental lung metastasis model by tail vein injection. For each group, six mice were used for quantification. **b, d** Representative pictures and quantitative results of hematoxylin and eosin (H&E) staining of lung or liver tissue sections from indicated mice in **a** and **c**. Left panel: H&E images; right panel: the quantitative data for lung metastatic foci coverage. Scale bar, 1 mm. **e** Luciferase-based bioluminescence imaging on the liver and lung of indicated mice treated with EVs derived from ITGBL1-overexpressing or -silent CRC cells relative to their respective control cells, or blank controls, or NCM-460. Luciferase-labeled HCT116 cells were used to perform liver metastasis model by cecum orthotopic transplantation. For each group, six mice were used for quantification. **f** Representative pictures and quantitative results of H&E staining of liver tissue sections from indicated mice in **e**. Left panel: H&E images; right panel: the quantitative data for liver metastatic foci coverage. Scale bar, 1 mm. **g** Immunofluorescence and quantitative analysis of fibroblast markers (α-SMA) in experimental lung metastatic tissues, experimental liver metastatic tissues, and cecum orthotopic liver metastatic tissues from indicated mice in **a**, **c** and **e**. Scale bar, 100 μm. **h** Indicated gene (IL-6, IL-8, IL-1β, α-SMA, TGF-β, and CXCL12) expressions in the CAFs isolated from lung metastasis in **a** were detected by qRT-PCR analysis. **i** Indicated gene expressions (IL-6, IL-8, IL-1β, α-SMA, TGF-β, and CXCL12) in the CAFs isolated from liver metastasis in **b**. **j** Indicated gene expressions (IL-6, IL-8, IL-1β, α-SMA, TGF-β, and CXCL12) in the CAFs isolated from liver metastasis in **c**. Each experiment was performed at least in triplicate and the results are shown as mean ± SD. Student's t-test was used to analyze the data; *p < 0.05, **p < 0.01.

mediate cancer metastasis[13,14,38] and drug resistance[39,40]. Our current study for the first demonstrates high ITGBL1-containing EVs facilitate CRC metastatic growth in vivo. It is known that primary tumors produce factors and cytokines to affect metastatic tumor growth. However, what is not known is about how primary tumors affect metastatic cancer growth through regulation of intracellular signaling. We show that primary tumor-derived EVs contain docking mechanisms for specific cell types in distal organs, a process for preparing metastatic niche formation. ITGBL1-activated fibroblasts in distal organs facilitate metastatic tumor growth through several mechanisms: (1) activated fibroblasts serve as feeder cells; (2) activated fibroblasts release tumor growth-promoting factors and cytokines; (3) activated fibroblasts produce matrix proteins to support tumor growth; (4) activated fibroblasts produce proteases necessary for tumor mass expansion; and (5) activated fibroblasts communicated with other stromal cells, such as inflammatory cells to induce tumor growth.

RUNX2, the upstream transcriptional regulator of ITGBL1, is regarded as a pivotal transcription factor for the formation of the osteomimetic phenotype[41,42] and contributes to cancer cell dissemination into the blood, survival in the bone microenvironment, and stimulation of bone resorption[43,44]. In prostate cancer, RUNX2 plays a significant role in intratibial prostate cancer-related tumor growth and bone loss[45]. In low-metastatic potential MCF-7 cells, high levels of RUNX2 significantly enhanced cancer invasiveness[46]. In the present study, we show that RUNX2 drives ITGBL1 transcription and induce secretion of the ITGBL1-loaded EVs through a nSMase2-mediated ceramide-dependent secretion pathway in CRC cells. Additionally, RUNX2 promotes the CRC EMT through activation of fibroblasts, i.e., the transition into myofibroblasts, which are known to facilitate tumor growth and invasiveness (Supplementary Fig. 19).

Therapeutic strategies that target the tumor stromal components in the TME are crucial for inhibiting tumor metastasis[47]. CAFs, one of the most abundant cell types in the tumor stroma, play important roles in promoting tumor progression. CAFs modulate the inflammatory microenvironment by expressing pro-inflammatory cytokines[48,49]. The crosstalk between tumor cells and CAFs has been studied extensively[50,51]. In our study, we found that tumor-derived ITGBL1-enriched EVs are directly transferred to lung and liver metastatic niche, and converts fibroblasts and stellate cells to CAFs by binding to TNFAIP3, and activating the NF-κB signaling pathway. Moreover, CAFs secrete pro-inflammatory cytokines, such as IL-6 and IL-8 to promote tumor progression[29]. A recent study shows the crucial roles of IL-6 and hepatocytes in the formation of pre-metastatic niche in the liver[12]. The intercommunication between tumor cells and CAFs further elucidated the molecular mechanism of CRC metastatic growth in liver and lung, and explained why diverse

types of CRC show different metastatic and growth abilities in the liver and lung.

Our findings also demonstrate important roles of the circulating tumor-derived EVs and their contents in activating CAFs and promoting CRC metastatic growth. In a previous report, integrins α6β4 and αvβ5 in EVs emerge as two predictors for organ-specific metastasis[18]. We show that ITGα6, ITGβ4, and ITGβ1 levels are high in the plasma EVs of CRC patients and correlate with lung metastasis, whereas high ITGα5 and ITGβ5 correlate with liver metastasis in CRC patients. Our findings show that EVs mediate a more broad metastasis in various organs, especially in lung and liver. Therefore, therapeutic targeting the EVs-mediated metastasis would have greater benefits for CRC patients and this concept warrants future clinical validation.

Activated CAFs secrete an array of pro-inflammatory cytokines, such as IL-6, IL-8, and IL-1β. They also produce high levels of TGF-β, α-SMA, and CXCL12[27–29] to drive toward a myofibroblast direction. Due to the reason that the typical of myofibroblastic CAFs activity in many cases is performed by a different subpopulation of CAFs[52], the upregulation of TGF-β and α-SMA in our study also demonstrated that EVs education activated not only pro-inflammatory function but also wound-healing functions in fibroblasts. Interestingly, our preliminary study also showed that TGF-β in the EVs was not alerted in ITGBL1 overexpressing or silencing CRC cells (Supplementary Fig. 20a), and their effect on TGF-β/Smads signaling pathway was little in the targeted HCT116 cells and CAFs from liver metastatic tumors (Supplementary Fig. 20b, c). Nevertheless, we observed the increased secretion of cytokines TGF-β in the activated hepatic stellate cells LX-2 upon indicated EVs treatment (Supplementary Fig. 20d), companying with positive effect on TGF-β/Smads signaling pathway in HCT116 cells (Supplementary Fig. 20e, f). Their functional impact on CRC progression warrants future investigation.

Using immunedeficient mice, we demonstrate the ITGBL1-enriched EVs derived from highly metastatic cancer cells accelerate metastatic cancer growth through a CAF activation mechanism. However, the using model of the athymic nude mice has certain limitations. Deficiency of T lymphocytes considerably immunocompromises the mice and enables the engraftment, growth, and eventually metastasizing of the tumor cells from the subcutaneous or orthotopic tumor xenograft[53]. Nevertheless, the athymic nude mice lack the proper T lymphocytes immune response at the site of primary tumor[54], which could not completely reflect the clinical scenario of primary tumor[55]. Therefore, considering the effect of host immune system, we explored the corresponding biological function and mechanism in immunocompetent C57Bl/6 mice. Consistent with the results in

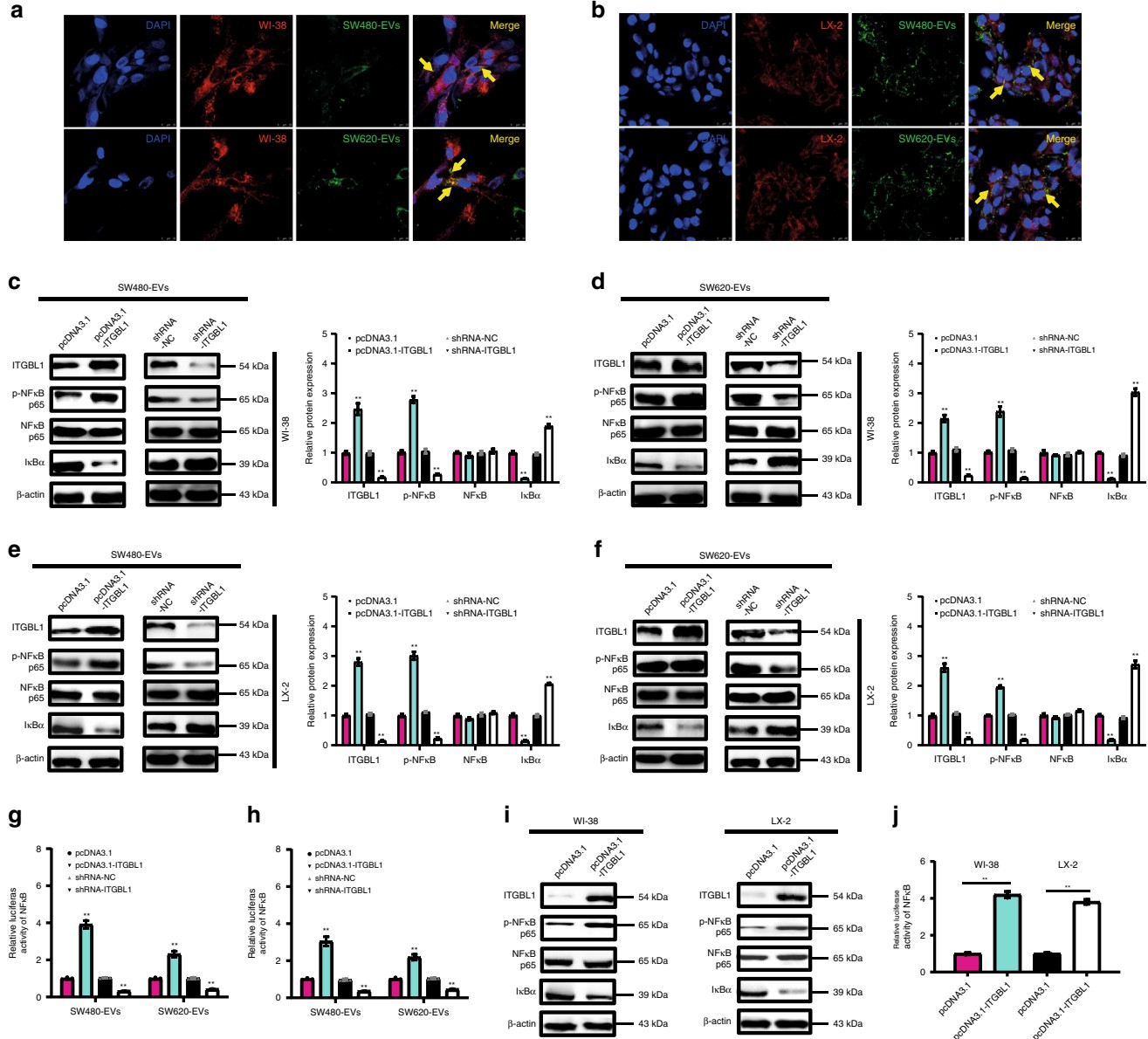

**Fig. 5 ITGBL1-enriched EVs activated fibroblasts via NF-κB signaling axis. a** Confocal imaging showed the delivery of PKH67-labeled EVs (green) to PKH26-labeled WI-38 (red). EVs were derived from SW480 or SW620 cells. Yellow arrows represented delivered EVs in the representative images. Scale bar, 25 μm. **b** Confocal imaging showed the delivery of PKH67-labeled EVs (green) to PKH26-labeled LX-2 (red). EVs were derived from SW480 or SW620 cells. Yellow arrows represented delivered EVs in the representative images. Scale bar, 25 μm. **c** Immunoblotting and quantitative assays of indicated proteins in WI-38 cells treated with EVs derived from ITGBL1-overexpressing or -silent CRC SW480 cells, or respective controls. **d** Immunoblotting and quantitative assays of indicated proteins in WI-38 cells treated with EVs derived from ITGBL1-overexpressing or -silent CRC SW620 cells, or respective controls. **e** Immunoblotting and quantitative assays of indicated proteins in LX-2 cells treated with EVs derived from ITGBL1-overexpressing or -silent CRC SW480 cells, or respective controls. **f** Immunoblotting and quantitative assays of indicated proteins in LX-2 cells treated with EVs derived from ITGBL1-overexpressing or -silent CRC SW620 cells, or respective controls. **g**, **h** Relative luciferase activity of NF-κB in WI-38 or LX-2 cells treated with EVs derived from ITGBL1-overexpressing or -silent CRC cells, or respective controls. **i** Immunoblotting assays of indicated proteins in ITGBL1-overexpressing WI-38 and LX-2 cells, or control cells. **j** Relative luciferase activity of NF-κB in ITGBL1-overexpressing WI-38 and LX-2 cells, or control cells. Each experiment was performed at least in triplicate and all the data are shown as mean ± SD. Student's $t$-test was used to analyze the data; $*p < 0.05$, $**p < 0.01$.

immunedeficient nude mice, education of CRC-ITGBL1-enriched EVs increased recruitment of α-SMA + hStCs, myofibroblasts, FN deposition, F4/80[+] macrophage, and Ly6G[+] myeloid cells migration to the liver and lung of C57Bl/6 mice (Supplementary Fig. 21a, b), supporting the concept of a pre-metastatic niche formation. Importantly, the experimental lung and liver metastasis model, and the cecum orthotopic tumor metastasis model validate ITGBL1-enriched EVs accelerated metastatic growth (Supplementary Fig. 22a–c). Expression levels of *IL-6*, *IL-8*, *IL-1β*,

*α-SMA*, *TGF-β*, and *CXCL12* are increased in ITGBL1-enriched EVs-treated CAFs (Supplementary Fig. 22d–f). The results from the immunocompetent C57Bl/6 mice validate those results from the immunodeficient nude mice. Owing to the existence of autoimmune regulation in C57Bl/6 mice, the regulatory effect of ITGBL1-enriched EVs on EVs incorporation, EVs fusion with recipient cells, the pre-metastatic niche formation, and tumor metastatic growth may not be identical and warrants future investigation.

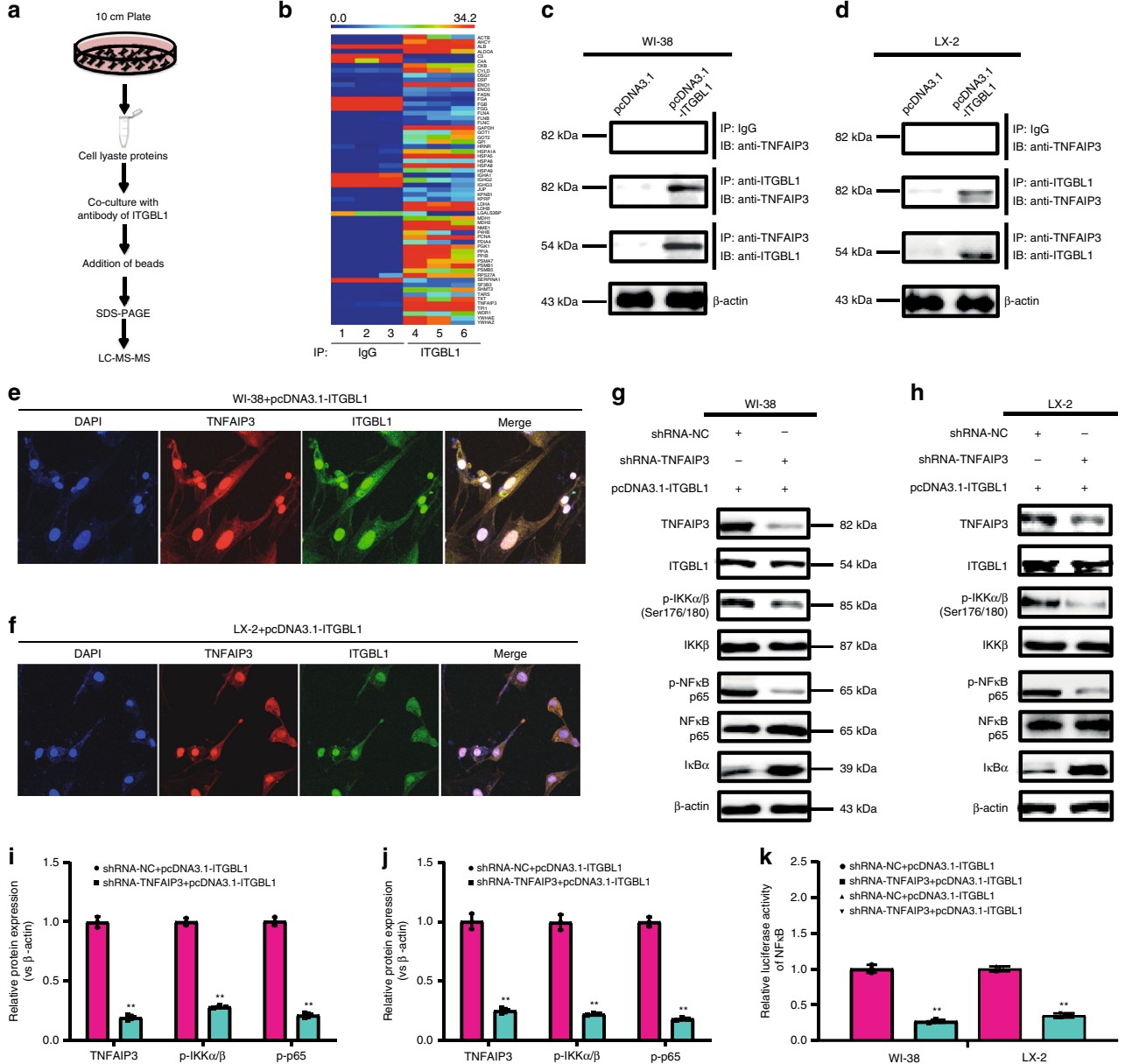

**Fig. 6 TNFAIP3 was a direct binding partner of ITGBL1 in NF-κB signal-mediated fibroblast activation. a** The schematic procedure of probing the binding partners of ITGBL1 in fibroblasts by Co-IP and LC-MS/MS analysis methods. **b** Co-IP, LC-MS/MS, and cluster analysis of the binding proteins for targeted ITGBL1 protein using the ITGBL1 antibody (samples 4, 5, and 6) or control IgG (sample 1, 2, and 3) in ITGBL1-overexpressing WI-38 cells. Red represents high scores and green represents low scores. The color brightness of each unit is associated with differences in multiples (log 2[AR/N]). **c, d** Co-IP in combination with western blot performed to validate the interaction between ITGBL1 protein and TNFAIP3 protein in ITGBL1-overexpressing WI-38 and LX-2 cells, or control cells. **e, f** Immunofluorescence detection of the co-location of ITGBL1 and TNFAIP3 in ITGBL1-overexpressing WI-38 and LX-2 cells, or control cells. Scale bar, 50 μm. **g–j** Western blot and quantitative assay of TNFAIP3, p-IKKα/β, and p-p65 in TNFAIP3-silent or/and ITGBL1-overexpressing WI-38 and LX-2 cells, or control cells. Figure 4i, j was the quantitative results for Fig. 4g, h, respectively. **k** Relative luciferase activity of NF-κB in TNFAIP3-silent or ITGBL1-overexpressing WI-38 and LX-2 cells, or control cells. Each experiment was performed at least in triplicate and all the data are shown as mean ± SD. Student's $t$-test was used to analyze the data; $*p < 0.05$, $**p < 0.01$.

In conclusion, our data provide a concept and paradigm to support the EVs-mediated cancer metastatic niche formation in distal organs. Therapeutic interference of this pathway, especially the RUNX2-regulated ITGBL1-loaded EVs, provides an attractive approach for treating CRCs. Although we use CRC as an example, this therapeutic paradigm may also apply to other cancer types.

## Methods

**Human tissues and cell lines**. A total of 78 human primary CRC tissues without metastasis and 46 human primary CRC tissues with matched hepatic or lung metastasis were collected between 2006 and 2015 at Shuguang Hospital, Shanghai University of Traditional Chinese Medicine, and Fudan University Shanghai Cancer Center. The general clinical information of all the 124 CRC patients was included in Supplementary Table 4. The serum samples from 32 healthy volunteers were used as control. HEK293T cell line, human CRC cell lines HCT116, Caco2, LoVo, SW480, SW620, human lung fibroblasts WI-38, and human hepatic stellate cell line LX-2 were purchased from American Type Culture Collection. HEK293T, HCT116, Caco2, LoVo, SW480, and SW620 were cultured in RPMI1640, Dulbecco's Modified Eagle Medium (DMEM), F12K medium, or Leibovitz's L-15 medium. WI-38 cells and LX-2 cells were cultured in alpha-MEM and DMEM medium, respectively. Normal colonic epithelial cells NCM-460 (cultured in DMEM) was used as control. All procedures were conducted with the approval of

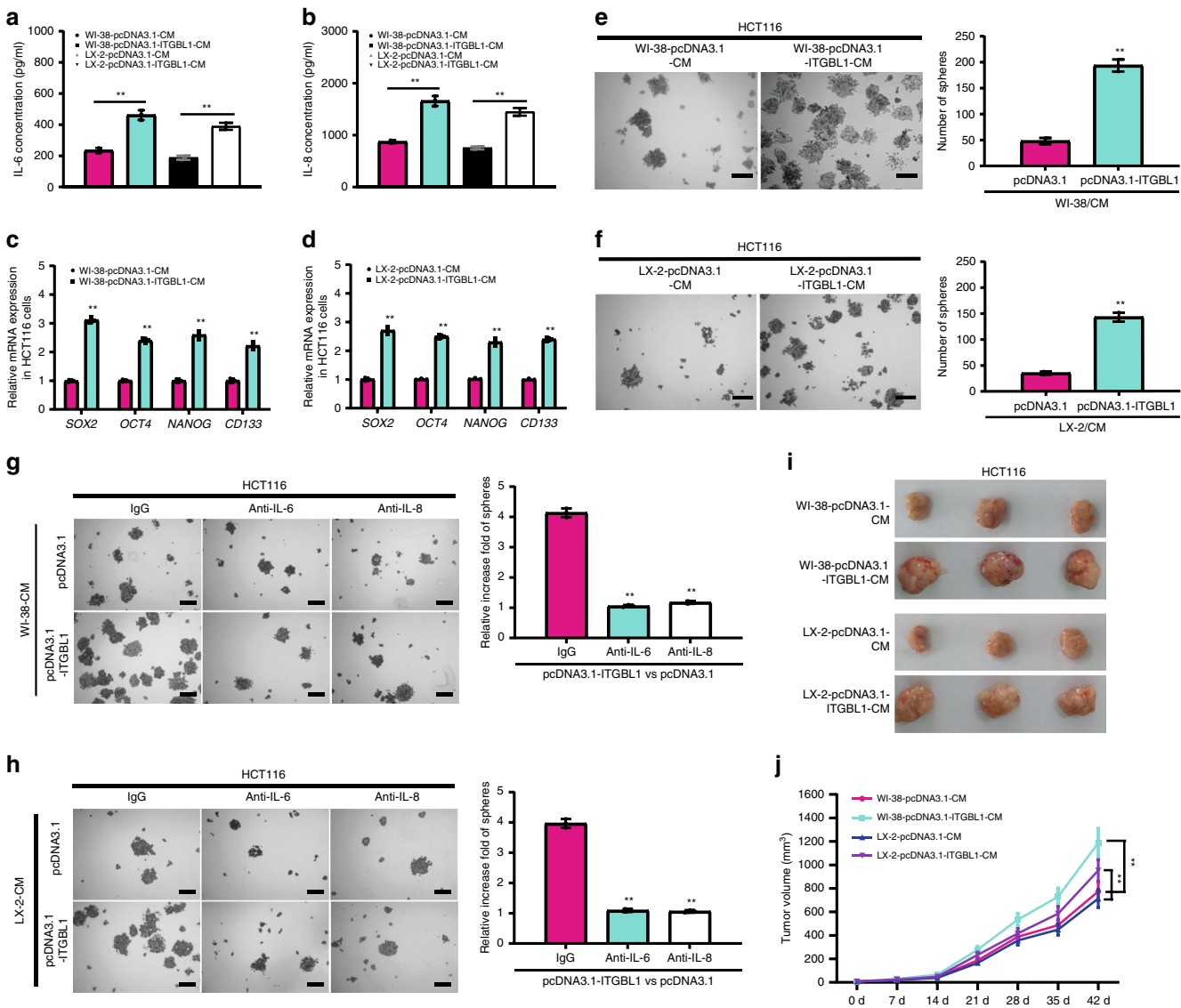

**Fig. 7 Activated fibroblasts by ITGBL1-enriched EVs promoted the stemness and tumorigenicity of CRC. a, b** ELISA assays of the IL-6 and IL-8 secretion from ITGBL1-overexpressing WI-38 and LX-2 cells, or control cells. **c, d** qRT-PCR analysis of stemness-associated genes expression in HCT116 cells treated with indicated conditioned medium (CM) from ITGBL1-overexpressing WI-38 and LX-2 cells, or control cells. **e, f** Spheroid formation ability of HCT116 cells treated with indicated CM from ITGBL1-overexpressing WI-38 and LX-2 cells, or control cells. Representative images were shown and spheroids were counted. Scale bar, 150 μm. **g, h** Relative spheroid formation ability of HCT116 cells treated with indicated CM containing anti-IL-6 antibody or anti-IL-8 antibody or control IgG antibody. Representative images and relative quantification were shown. Scale bar, 150 μm. **i, j** Xenografts of HCT116 cells with indicated treatments performed on nude mice. HCT116 cells were pretreated with the CM from ITGBL1-overexpressing WI-38 and LX-2 cells, or control cells. Length and width of the subcutaneous tumors were measured every 7 days. For each group, three mice were used for quantification. Each experiment was performed at least in triplicate and all the data are shown as mean ± SD. Student's $t$-test was used to analyze the data; $*p < 0.05$, $**p < 0.01$.

the Ethical Committee of Shuguang Hospital, Shanghai University of Traditional Chinese Medicine, and Fudan University Shanghai Cancer Center. Written informed consent was obtained from all patients before the start of the study.

**RNA sequencing and bioinformatics analysis**. CRC frozen tissues (20 mg each) were disrupted and homogenized, following by RNAs isolation. For MACE (Massive Analysis of cDNA Ends), MACE libraries were firstly prepared[56]. Briefly, poly-adenylated RNA was extracted (Poly (A) mRNA Magnetic Isolation Module, NEB) from large RNA fractions and reverse transcribed (mRNA Second Strand Synthesis Module, NEB) with biotinylated poly (dT) primers. cDNA was fragmented (E210 Covaris instrument, Covaris Inc., USA) to a 150–600 bp size range. Biotinylated ends were captured by streptavidin beads (Streptavidin Magnetic Beads, NEB) and ligated to modified adapters (mRNA Library Prep Reagent Set for Illumina®). The MACE libraries were amplified by PCR (Q5 High Fidelity Hot Start 2X Master Mix, NEB), purified by SPRI beads (Agencourt AMPure XP, Beckman Coulter) and sequenced (HiSeq2000, Illumina).

For bioinformatics analysis of MACE data, all duplicate reads were removed from the raw datasets to remove any PCR bias. The preprocessing reads were aligned to the human genome with novoalign (http://www.novocraft.com). The RefSeq annotation track that includes mRNAs and lncRNAs (http://genome.ucsc.edu/cgi-bin/hgTables) was applied in annotations for genomic mapping positions. Normalization and test for differential expression were calculated as described for small RNAs. Those genes with false discovery rate (FDR) < 0.05 and |log2fc| > 1.6 were considered as differentially expressed genes. The functional classifications and pathway annotation of the DEPs (differentially expression proteins) were performed with the DAVID (database for annotation, visualization and integrated discovery) functional annotation tool[57] (http://david.abcc.ncifcrf.gov) using an enrichment cutoff of FDR < 0.05.

**TMA analysis**. The protein expression of ITGBL1 in CRC tumors and adjacent normal tissues was detected using our previous made TMAs, which containing 124 CRC primary tumors tissues and matched adjacent normal tissues. The TMAs were used for IHC analysis using ITGBL1 rabbit antibody (17484-1-AP, 1: 200). The composite scoring system[58] was applied for statistical analysis, in

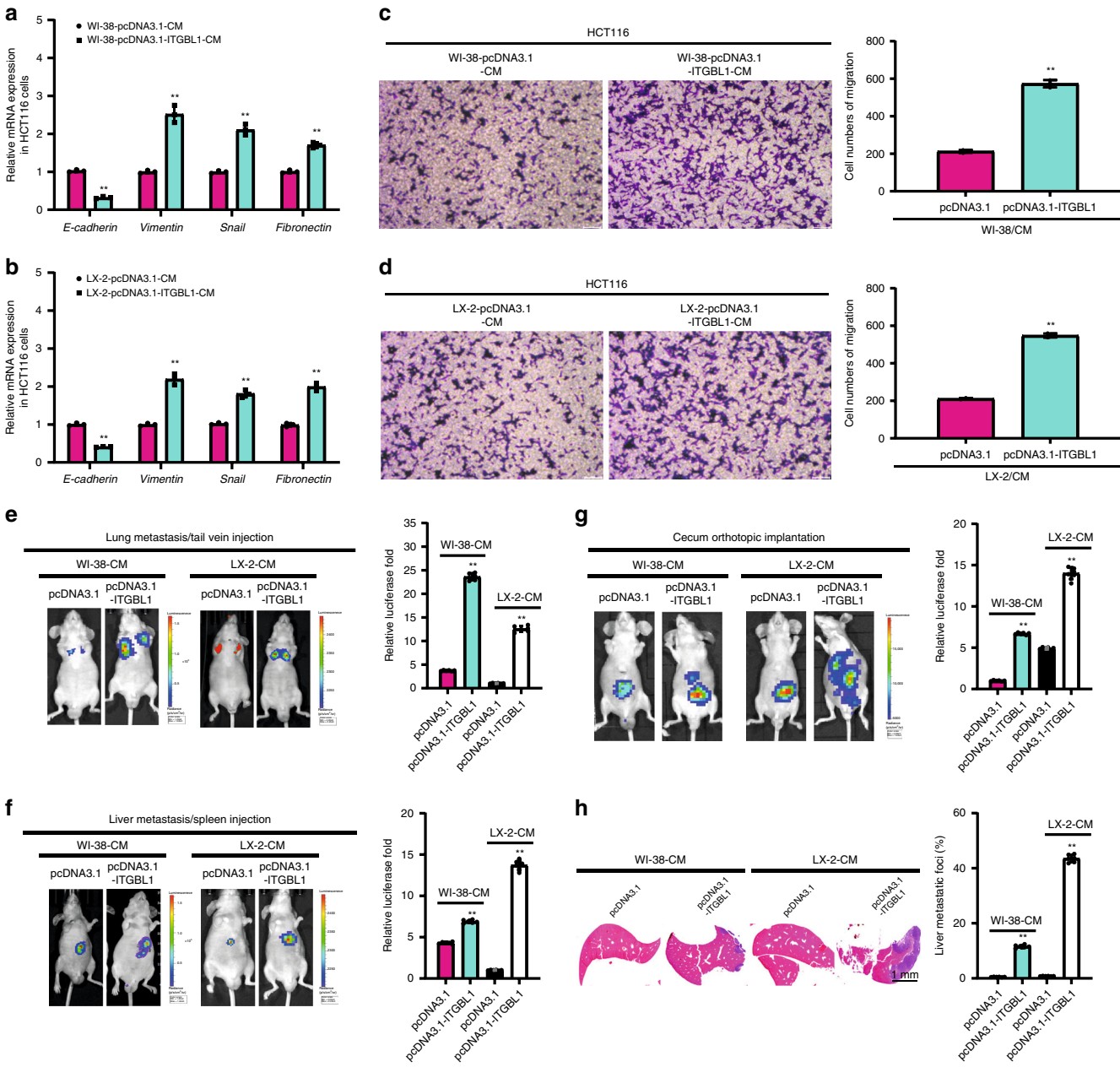

**Fig. 8 Activated fibroblasts by ITGBL1-enriched EVs promoted the EMT and migration of CRC. a, b** qRT-PCR analysis of EMT-associated gene expression in HCT116 cells treated with indicated conditioned medium (CM) from ITGBL1-overexpressing WI-38 and LX-2 cells, or control cells. **c, d** Migration assay of HCT116 cells treated with indicated CM from ITGBL1-overexpressing WI-38 and LX-2 cells, or control cells. Representative images are shown and migrated cells were counted. Scale bar, 150 μm. **e** Experimental lung metastasis assays of HCT116 cells with indicated treatments performed on nude mice. HCT116 cells were pretreated with the CM treated with indicated CM from ITGBL1-overexpressing WI-38 and LX-2 cells, or control cells. Representative images and quantitative analysis of lung metastases of indicated mice were determined by luciferase-based bioluminescence imaging. For each group, six mice were used for quantification. **f** Experimental liver metastasis assays of HCT116 cells with indicated treatments performed on nude mice. HCT116 cells were pretreated with the CM from ITGBL1-overexpressing WI-38 and LX-2 cells, or control cells. Representative images and quantitative analysis of liver metastases of indicated mice were determined by luciferase-based bioluminescence imaging. For each group, six mice were used for quantification. **g** Luciferase-based bioluminescence imaging on cecum orthotopic transplantation of indicated mice using luciferase-labeled HCT116 cells pretreated with the CM from ITGBL1-overexpressing WI-38 and LX-2 cells, or control cells. For each group, six mice were used for quantification. **h** Representative pictures and quantitative results of hematoxylin and eosin (H&E) staining of liver tissue sections from indicated mice in **g**. Left panel: H&E images; right panel: the quantitative data for liver metastatic foci coverage. Each experiment was performed at least in triplicate and all the data are shown as mean ± SD. Student's $t$-test was used to analyze the data; *$p < 0.05$, **$p < 0.01$.

which the system integrates the IHC signal intensity and the frequency of positive cells in the cytosol and nucleus. The immunoreactivity for each sample was observed and evaluated by two experienced pathologists. The IHC scores for intensity are between 0 and 3, and the IHC scores for frequency are between 0 and 4. A composite expression score (CES) was calculated using the following formula CES = 4 × (intensity − 1) + frequency.

**EVs purification and characterization**. The minimal experimental requirements for definition of EVs refer to the publication by Lötvall J, et al.[59]. All the EVs used in this study were purified by sequential centrifugation[14]. In brief, live cells, possible apoptotic bodies and large cell debris were removed from CRC patients' plasma or cell culture medium by two step centrifugation: $500 \times g$ for 10 min, and $12,000 \times g$ for 20 min. Then, the pellets containing EVs were collected by spinning

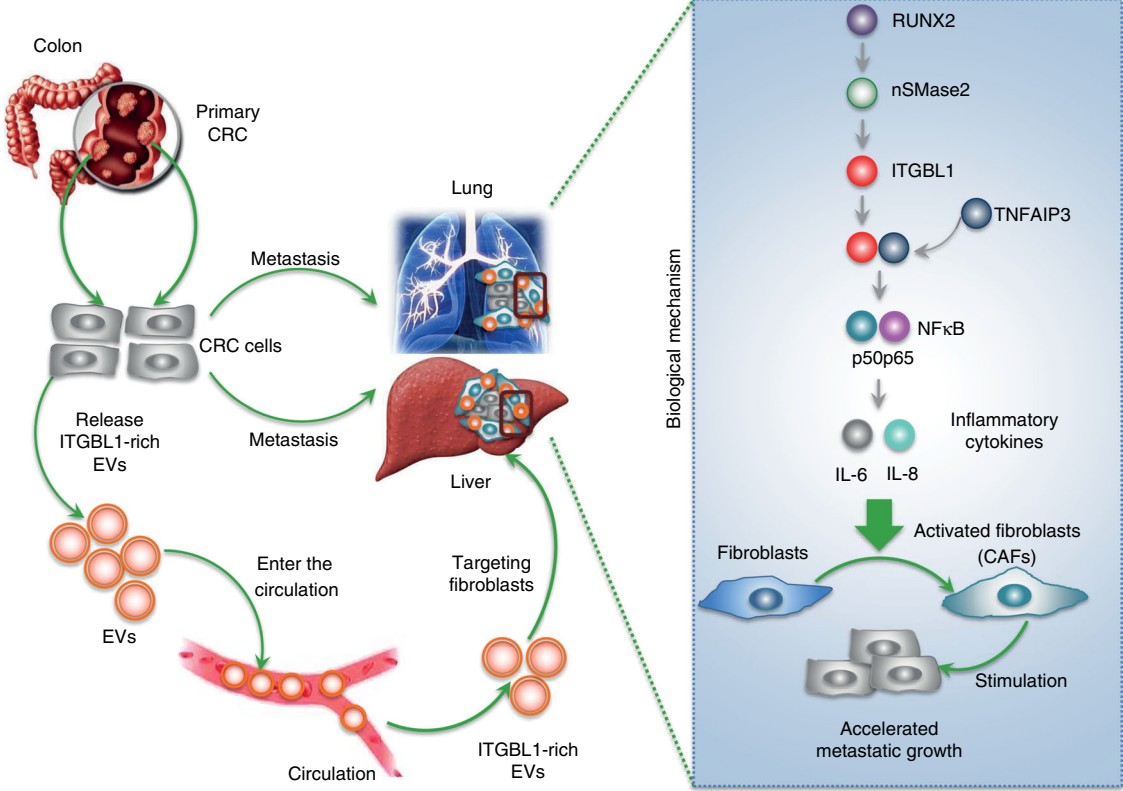

**Fig. 9 A schematic mechanism underlying interactions between primary CRC cells and distal fibroblasts through the EVs-mediated transfer of ITGBL1 in promoting the metastasic niche formation in lung and liver.** In detail, for biological function description, tumor-derived ITGBL1-enriched EVs convert primary fibroblasts to CAFs via binding to TNFAIP3 and activating the NF-κB signaling pathway. The ITGBL1-activated CAFs promote CRC stemness, aggressiveness, and EMT by pro-inflammatory cytokines, such as IL-6 and IL-8. CAFs manipulated the TME to support metastatic tumor growth. For the molecular mechanism of ITGBL1 secretion, in primary CRC, the transcription of ITGBL1 was regulated by RUNX2, and the secretion of ITGBL1-loaded EVs was elevated by RUNX2 through the ceramide-dependent EVs secretion pathways mediated by nSMase2.

at 100,000 × g for 70 min. After washing with phosphate-buffered saline (PBS), the pellets were deal with ultracentrifugation (Beckman 70Ti rotor). The EVs quality, size, and particle number were characterized by electron microscopy and LM10 nanoparticle characterization system (NanoSight, Malvern Instruments). All the purified EVs pellets were resuspended in PBS and the protein concentration was determined by bicinchoninic acid (BCA; Pierce, Thermo Fisher Scientific). Normal EVs from healthy volunteers' plasma or normal colonic epithelial cells NCM-460 were also purified and characterized for the control in our studies.

**Construction of the expression vectors**. Human *ITGBL1*, *RUNX2*, and *nSMase2* genes were amplified by reverse transcription polymerase chain reaction (RT-PCR) and sub-cloned into the expression vector pcDNA3.1, and were named pcDNA3.1-ITGBL1, pcDNA3.1-RUNX2, and pcDNA3.1-nSMase2, respectively (Supplementary Table 5). The Ubi-Luc-MCS plasmid was purchased for lung and liver metastases mice models, and in vivo optical imaging. For constructing the *ITGBL1* promoter luciferase reporter with or without the RUNX2-binding element −793 to −786 (ACCACAAA) relative to the transcription start site (TSS), the *ITGBL1* promoter regions −825 to 36 were amplified and inserted into the pRF-basic vector, and this was named pRF-ITGBL1 (−825/+36). pRF-ITGBL1 (−825/+36) with a mutated RUNX2-binding motif (ACCTCAAA) was constructed for negative control.

**Cell transfection and lentivirus infection**. Transient transfections were performed using the Lipofectamine 2000 kit (Invitrogen) according to the manufacturer's instructions. Recombinant lentiviruses containing pLV4-shRNA-ITGBL1, pLV4-shRNA-RUNX2, pLV4-shRNA-nSMase2, pLV4-shRNA-TNFAIP3, pLV4-shRNA-ITGβ1, and pLV4-shRNA-NC (non-targeting control) were constructed, and the suitable shRNA sequences were listed in Supplementary Table 6.

**Luciferase reporter assay**. Approximately 8000 HEK293T cells per well or 12,000 LoVo cells per well were plated into 96-well plates and were co-transfected with 50 ng of the Firefly luciferase reporter and 5 ng of the pRLCMV Renilla luciferase reporter using Lipofectamine 2000 kit (Invitrogen). After incubating for 48 h, the Firefly and Renilla luciferase activities were quantified using a dual-luciferase

reporter assay (Promega). Data were presented as the relative ratio of Firefly luciferase activity to Renilla luciferase activity.

**Quantitative real-time PCR**. Total RNA was extracted using the TRIzol reagent (Invitrogen) according to the manufacturer's instructions. The concentrations of RNA were determined using a NanoDrop ND-1000 (NanoDrop), and aliquots of the samples were stored at −80 °C. cDNA was synthesized with the PrimeScript RT Reagent Kit (TaKaRa) using 500 ng total RNA as template. Quantitative PCR (qPCR) analyses were conducted to quantitate mRNA relative expression using SYBR Premix Ex Taq (TaKaRa) with GAPDH as an internal control. The results of qPCR were defined from the threshold cycle (Ct), and relative expression levels were calculated by using the $2^{-\triangle\triangle Ct}$ method. PCR was performed using an ABI 7500 instrument (Applied Biosystems). The primers used for reverse transcription and PCR analysis were listed in the Supplementary Table 7.

**Chromatin immunoprecipitation**. ChIP assays were performed using a ChIP assay kit (Millipore, USA). Anti-GST tag antibody (2625, 1:200) was used to enrich the *ITGBL1* promoter fragments containing the putative RUNX2-binding sites in RUNX2-GST-overexpressing CRC cells. The nonspecific mouse IgG antibody (ab6709, 1:100) was used as a negative control. The primers for the amplification of the *ITGBL1* promoter regions −825 to +36 containing the putative OSE2 element −793/−786 relative to the TSS were listed in Supplementary Table 7. Uncropped versions of gels shown in figures are supplied in the Supplementary Information.

**LC-MS/MS analysis**. MS analyses were performed using 20 μg proteins from Co-IP assays. Samples were denatured using 8 M urea, reduced using 10 mM dithiothreitol, and alkylated with 100 mM iodoacetamide. Then the samples were proteolytic digested with endoproteinase LysC overnight at room temperature, following by digestion with trypsin (Promega) for 15 h at 37 °C. The resulting peptide mixtures were extracted using peptide extract solution (50% ACN (acetonitrile), 0.1% TFA (trifluoroacetic acid)) for 30 min at 37 °C. This protocol was repeated twice. Then the samples were dried and solubilized in the sample loading buffer containing 0.1% formic acid. Each sample of ~3–5 μg was analyzed by reversed phase nano-LC-MS/MS (Thermo Scientific). LC-MS/MS data from three technical replicates of two Co-IP samples were analyzed using MaxQuant and

Perseus software by searching the Uniprot human database[18]. Label-free quantitative values represent the protein abundance. The FDR values at the protein and peptide levels were set to 1%. Only those proteins quantified in at least two out of three replicates in at least one group were reserved, and the missing values were imputed. The multiple sample analysis of variance (ANOVA) test was executed and corrected for multiple hypotheses testing using a cutoff of FDR < 0.05.

**Subcellular fractionation.** For the whole-cell proteins, the harvested cells were washed with cold PBS and then lysed in RIPA lysis buffer, and the lysates were cleared by centrifugation at $12,000 \times g$, 4 °C for 15 min. For cytosolic and nuclear fractions, cells were firstly lysed with a buffer containing 10 mM Tris-HCl, pH 7.4, 100 mM, NaCl, 2.5 mM $MgCl_2$, and 40 μg/ml digitonin for 15 min, and the lysates were centrifuged at $2060 \times g$ for 15 min at 4 °C. The supernatant was just the cytosolic fraction. Subsequently, the pellets were washed, incubated with RIPA buffer at 4 °C for 15 min, and the nuclear fraction was collected after centrifugation at 4 °C for 30 min at $21000 \times g$.

**Western blot.** The frozen CRC tissues and paired metastatic tissues, or adjacent normal tissues or harvested cells were washed with cold PBS and then lysed in RIPA lysis buffer, and the lysates were cleared by centrifugation at $12,000 \times g$, 4 °C for 15 min. Harvested proteins were first separated by 10% sodium dodecyl sulfate–polyacrylamide gel electrophoresis (SDS–PAGE) and then transferred to nitrocellulose membranes (Bio-Rad Laboratories). The membranes were blocked with 5% nonfat milk and incubated with a mouse or rabbit first antibody at a dilution of 1:500 to 1:1000 (CST, Abcam, Proteintech). The membranes were subsequently incubated with a goat anti-mouse (7076, 1:3000) or goat anti-rabbit (7074, 1:3000) horseradish peroxidase secondary antibody. The protein complex was detected using enhanced chemiluminescence reagents (Pierce). The antibodies against the following proteins were purchased for western blot analysis: ITGBL1 (17484-1-AP, 1:500), HSP70 (4876, 1:1000), CD63 (ab193349, 1:1000), CD9 (ab92726, 1:1000), RUNX2 (12556, 1:1000), nSMase2 (ab68735, 1:500), p-NFκB (3033, 1:1000), NFκB (3034, 1:1000), IκBα (4812, 1:1000), TNFAIP3 (5630, 1:1000), p-IKKα/β (2697, 1:1000), IKKβ (8943, 1:1000), ITGα6 (3750, 1:1000), ITGβ4 (4707, 1:500), ITGβ1 (4706, 1:1000), ITGα5 (ab117611, 1:500), ITGβ5 (4708, 1:500), E-cadherin (3195, 1:1000), Vimentin (5741, 1:1000), Smad2 (5339, 1:500), p-Smad2 (3108, 1:500), Smad3 (9523, 1:500), and p-Smad3 (9520, 1:500). β-Actin (ab179467, 1:5000) was used as the internal control. For protein detection in EVs, the extracted EVs were suspended in SDS lysis buffer and quantified by BCA (Pierce, Thermo Fisher Scientific), and the following procedure was as above. Uncropped versions of all blots shown in main figures are supplied in the Supplementary Fig. 23–33.

**EVs treatment and labeling.** To assess lung, liver, brain, and bone marrow EVs distribution, the purified EVs were injected via the retro-orbital venous sinus according to the protocols provided by Hoshino et al.[18]. For 24 h EVs treatments, 10 μg (in 100 μl PBS) of total EVs protein were injected via the retro-orbital venous sinus. To measure EVs uptake by specific cell types, labeled EVs were injected 24 h before tissue collection and tissues were analyzed for EVs-positive cells by immunofluorescence.

For in vitro uptake assays, the membrane of WI-38 or LX-2 cells was labeled with PKH67 dye, while the EVs were labeled with PKH26 dye. The labeled EVs were washed in 20 ml of PBS, ultracentrifuged, and then resuspended in PBS. EVs (10 μg in 1 ml) were first incubated with WI-38 or LX-2 cells for 1 h at 37 °C. Excess EVs were washed off by PBS. Then, the cells were fixed with 4% paraformaldehyde (PFA) for 20 min. Nuclear staining was performed with DAPI (40, 6-diamidino-2-phenylindole). The pictures were taken by a TCS SP2 spectral confocal system (Leica, Ernst-Leitz, Wetzlar, Germany).

**Tumor mouse model.** For EVs localization, education, and tumor implantation experiments using human cell lines, 6–8-week-old female nude mice were used. For in vivo EVs education administration, 6–8-week-old female nude mice received 10 μg of EVs retro-orbitally every other day for 3 weeks. Experimental lung metastases were induced by injections of a Luc-labeled HCT116 single-cell suspension ($2 \times 10^6$ cells in 100 μl) into the mouse lateral tail vein[18]. Experimental liver metastases were achieved by intra-splenic injection of a Luc-labeled HCT116 single-cell suspension at $2 \times 10^6$ cells/injection site[7]. For cecum orthotopic implantation and metastasis models[60,61], Luc-labeled HCT116 single-cell suspension ($2 \times 10^6$ cells in 100 μl) was injected into the mice cecum. Five to 7 weeks later, prior to in vivo imaging, the mice were anesthetized with phenobarbital sodium, and established lung and liver metastases images were observed by LB983 NIGHTOWL II system (Berthold Technologies GmbH, Calmbacher, Germany). Next, the mice were anesthetized with phenobarbital sodium, the lung, liver, brain, and bone organs were excised, and the metastatic lesions were determined by hematoxylin and eosin (H&E) staining, and IHC staining. All experimental procedures involving mice were carried out as prescribed by the National Guidelines for Animal Usage in Research (China) and were approved by the Ethics Committee of Shuguang Hospital, Shanghai University of Traditional Chinese Medicine.

**Isolation and primary culture of CAFs.** The primary CAFs or NFs were isolated from fresh liver metastatic tumor tissues or normal liver tissues[62,63]. Briefly, tumor tissues were cut into small pieces of 1 mm$^3$ using a razor blade, and the tissue pieces were dissociated using the Tumor Dissociation Kit (Miltenyi Biotec) according to the manufacturer's procedure. Cells were then resuspended, passed through a cell strainer (100 μM), and finally plated into a T75 flask. Tissue blocks trapped in the cell strainer were seeded in 10 cm$^2$ culture dishes in order to isolate more CAFs by outgrowth. Cells were cultured in DMEM or F12 medium (Gibco) supplemented with 10% fetal bovine serum (FBS, Gibco), 2 mmol/L glutamine (Invitrogen), 0.5% sodium pyruvate (Invitrogen), and 1% antibiotic-antimycotic (Invitrogen). The primary CAFs were characterized by immunofluorescence using a positive α-SMA and FAP (fibroblast activation protein) staining, and a negative EpCAM, CD45, and KRT19 staining. The first to sixth passages of the primary fibroblasts were used in our experiments. At each passage, β-galactosidase staining (Cell Signaling) was performed according to the manufacturer's recommendations, to ensure cells were not in senescence.

**Protein immunoprecipitation.** Cells of each group were lysed in lysis buffer containing 150 mM NaCl, 100 mM Tris-HCl (pH 7.4), 10% v/v glycerol, 0.5% Triton X-100, and protease inhibitor cocktail. Cell lysates were then centrifuged and the supernatants were obtained. Protein A/G sepharose beads were added to the supernatants to preclear the nonspecific binding. Then the primary antibody was added and incubated with precleared lysates at 4 °C. After incubation for 24 h, protein A/G sepharose beads were added into the incubated mixture for 1 h. Next, the pellets were washed three times with PBS buffer and eluted with SDS–PAGE sample buffer, following by western blot detection with secondary antibody and the primary antibody. All the assays were performed in triplicate and independently repeated at least three times.

**Transwell analysis.** A total of $2.5 \times 10^5$ cells (in 100 μL F12K with 0.5% fetal bovine serum) were seeded into the upper part of a transwell chamber (transwell filter inserts in 6.5 mm diameter, pore size of 8 μm, Corning Incorporated, NY, USA). In the lower part of the chamber, 600 μL F12K with 15% FCS and 10 μg/ml FN was added and the assay was performed after 24 h at 37 °C and 5% $CO_2$. Migrated cells were analyzed by crystal violet staining, followed by observation under a DMI3000 B inverted microscope (Leica, Ernst-Leitz, Wetzlar, Germany). Anti-IL-6 (AF-206) and anti-IL-8-neutralizing (MAB208) antibody were purchased from R&D Systems (Minneapolis, USA). All assays were performed in triplicate and independently repeated three times.

**Subcutaneous transplantation tumor model.** For the subcutaneous transplantation tumor model, the EVs-educated CRC or control cells were harvested from culture flasks and transferred to serum-free PBS. Single-cell suspensions ($2 \times 10^6$ in 100 μl) were injected into the subcutaneous area of female BALB/c nude mice (4–6 weeks old) obtained from SLAC (SLAC Laboratory Lab, Shanghai, China). Tumor size was evaluated using a standard caliper measuring tumor length and width in a blinded manner and the tumor volume was calculated using the formula: length × width$^2$ × 0.52. All the animal experiments were carried out as prescribed by the National Guidelines for Animal Usage in Research (China) and were approved by the Ethics Committee of Shuguang Hospital, Shanghai University of Traditional Chinese Medicine.

**Tumor mouse model in C57Bl/6 mice.** For EVs education and tumor implantation experiments using mouse cell line MC-38, 6–8-week-old C57Bl/6 Mus musculus females were used. For in vivo EVs education administration, 6–8-week-old female C57Bl/6 mice received 10 μg of EVs retro-orbitally every other day for 3 weeks. For orthotopic implantation and metastasis models, Luc-labeled MC-38 single-cell suspension ($2 \times 10^6$ cells in 100 μl) was injected into the cecum of mice. Seven weeks later, the mice were anesthetized with phenobarbital sodium, and the lung and liver organs were excised for imaging by LB983 NIGHTOWL II system (Berthold Technologies GmbH, Calmbacher, Germany). All the animal experiments were carried out as prescribed by the National Guidelines for Animal Usage in Research (China) and were approved by the Ethics Committee of Shuguang Hospital, Shanghai University of Traditional Chinese Medicine.

**Immunofluorescence staining.** The cells were collected and fixed with methanol, blocked with 5% bovine serum albumin. Then, the cells were firstly stained with TNFAIP3 mouse antibody (ab13597, 1:200) followed by Cy3-conjugated goat anti-mouse IgG (ab97035, 1:100). After the cells were washed four times with PBS, the ITGBL1 rabbit antibody (17484-1-AP, 1:200) was added, followed by FITC-conjugated goat anti-rabbit IgG (ab6717, 1:1000). Nuclear staining was done with DAPI. Immunofluorescence images were taken with a DMI3000B inverted microscope (Leica, Ernst-Leitz, Wetzlar, Germany). All experiments were conducted according to instructions from the antibody manufacturer.

**Tissue immunofluorescence.** For histological analysis, dissected tissues were fixed in a mixture of 2% PFA and 20% sucrose solution for 24 h at room temperature, and then embedded in Tissue-tek O.C.T. (Optimum Cutting Temperature) (Electron Microscopy Sciences). Blocks were frozen in a dry ice and ethanol bath. For immunofluorescence, 6 μm O.C.T. tissue cryosections were stained with

antibodies against α-SMA (ab32575, 1:100), F4/80 (ab100790, 1:200), FN (66042-1-Ig, 1:100), Ly6G (ab25377, 1:100), CD31 (ab9498, 1:100), and S100A4 (16105-1-AP, 1:100). Secondary antibodies conjugated to Alexa Fluor 488 (A-11008, 1:2000) and Alexa Fluor 594 (A-11032, 1:1000) were used (Life Technologies). Nuclear staining was done with DAPI. Immunofluorescence images were taken with a DMI3000B inverted microscope (Leica, Ernst-Leitz, Wetzlar, Germany). All experiments were conducted according to instructions from the antibody manufacturer.

**IHC staining**. Tissue arrays were conducted using 124 CRC tissues and 46 paired metastatic tissues. IHC staining was performed on 4 mm sections of paraffin embedded tissues to determine the expression level of proteins. In brief, the slides were incubated in first antibody diluted 1:50 to 1:200 at 4 °C overnight. The subsequent steps were performed using the EnVision FLEX High pH visualization system according to the manufacturer's instructions (Dako, Denmark).

**Sphere formation**. For sphere formation, indicated cells were seeded on six-well plates in a density of 3000 per well and cultured in RPMI1640 medium supplemented with 10% FBS. After observation under microscope, the number of spheroids was counted and representative images were captured.

**Statistical analysis**. OS was calculated from the surgery date to the death date or the last follow-up data. DFS was defined from the surgery date to the date of local or distant recurrence or the last follow-up data. OS rates and DFS rates were calculated actuarially according to the Kaplan–Meier method using log-rank test. The differences between groups were estimated using the $\chi^2$ test, the Student $t$-test, the Mann–Whitney $U$ test, and the repeated measures ANOVA test. The relationships between two groups were explored by the Pearson correlation analysis. Multivariate Cox regression models were used to analyze the correlations between risk factors and survival outcomes. All the data were presented as the mean values ± SEM. A two-tailed $P$ value of 0.05 or less was considered statistically significant. SPSS 22.0 and Graphpad prism 8.0 were used for statistical analyses and scientific graphing.

## Data availability

The RNA-seq data have been deposited in the GEO database under accession code GSE144259. The TCGA data referenced during the study can be obtained from the TCGA datebase (https://portal.gdc.cancer.gov) and cBioPortal datebase (http://www.cbioportal.org). The mass spectrometry proteomics data have been deposited to the ProteomeXchange Consortium (http://proteomecentral.proteomexchange.org) via the iProX partner repository (Ma J., et al. (2019) iProX: an integrated proteome resource. Nucleic Acids Res, 47, D1211-D1217) with the dataset identifier PXD017182. All the source data supporting the findings of this study are available within the article and its Supplementary Information files and from the corresponding author upon reasonable request.

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

## Acknowledgements

This work was supported by National Science Foundation of China (81830120, 81520108031 to Q.L.; 81573749 to Q.J.; 81973651 to L.Z.; 81573805 to H.S.; 81904013 to L.Y.; 81774095 to Y.W.; 81572351 to G.X.C., 81871958 to G.X.C, and 81703885 to Z.W.), 3 year plan of action for innovation of traditional Chinese medicine in Shanghai (ZY2020-CCCX-2003-03 to Q.L.), and Key project of Shanghai Municipal Science and Technology Commission (16401970500 to Q.L.).

## Author contributions

Q.J., Q.L., and G.X.C. conceived and designed the study. Q.L. and Y.W. supervised the project. Q.J., L.Y., Y.F., R.J., N.L., and G.X.C. conducted the bioinformatics analysis of RNA sequencing and clinical data analysis. Q.J., Q.S., J.C., and X.W. conducted the isolation and characterization of EVs. Q.J., X.W., Q.S., R.L., and Y.Y.F. performed the immunofluorescence analysis. L.Y., X.W., and Q.S. performed IP and ChIP experiments. J.S., Z.W., G.C., and X.S. conducted LC-MS/MS experiments. X.W., Q.S., R.L., Z.W., and X.S. performed all the animal studies and pathology analysis. R.L. and R.J. contributed to the isolation and characterization of CAFs and NFs. Y.Y.F. and N.L. performed real-time PCR measurement. X.W., Q.S., X.S., R.L., G.C., and J.S. conducted plasmids construction, cell transfection, and western analysis. Q.J., H.S., Y.W., G.X.C., and Y.F. performed statistical analysis. Q.J., L.Z., H.S., J.C., Y.C., and Q.L. performed overall data interpretation and wrote the paper.

## Competing interests

The authors declare no competing interests.
