## [Peer Review File · Nature Communications]

Reviewers' comments:

Reviewer #1 (Remarks to the Author):

In the manuscript by Qing Ji et al., the authors demonstrate that Colorectal cancer exosomes expressing Integrin Beta-like 1 activate cancer-associated fibroblasts to promote pro-inflammatory cytokines and EMT programs.

The work is timely and extends our knowledge of the importance of tumor exosome proteins in the promotion of metastasis.

First, the authors need to include important controls, such as normal colon exosomes, in their studies throughout (for example Figure 1B).

The authors should not assume that CAFs from the lung and stellate cells from the liver are the primary cells uptaken tumor exosomes. For instance other studies in the liver show that Kupffer cells are the primary cells take up exosomes. The authors need to preform a biodistribution study to show that these cells are the critical players. Moreover, the authors use WI-38 cell in their studies, this cell is a fibroblast and are not CAFs. Thus, the authors need to isolate CAFs from primary and metastatic tumors and incorporate these cells in their analyses.

It is also not clear how integrin beta-like 1 compared to B1 integrin. To demonstrate beta-like 1 intern's role, the authors should also knockdown B1 integrin. Does B-like 1 integrin bind to other integrins? what ligands does it bind to, other ECM molecules?

The paper by A. Hoshino et al also demonstrated that $\alpha 6 \beta 1$ integrin is relevant for lung mets.

Do the authors see a premetastatic site with exosome injection.

These models are not necessarily organotropic but clearly are promoting metastasis to lung and liver. Can they promote metastasis to bone for instance as well or is it specific to lung and liver.

Can the authors provide histology in all metastatic tissues.

Details are often missing, as in Figure 1F, what is each sample specifically

Interpretation of data is often not visualized in the panels, as with the spheroid numbers seem not to alter in Anti-IL-6 and anti-IL-8 experiments in the panels.

Mechanistically, the connections are missing for instance how is Runx2 related to EMT here.

Reviewer #2 (Remarks to the Author):

In this study, Ji et al. show that colorectal cancer cell lines secrete extracellular vesicles (EVs) containing high levels of integrin beta-like 1 (ITGBL1). Furthermore, these findings are nicely validated in human data from CRC patients and correlated with metastatic disease and with survival. CRC-derived EVs activated human lung fibroblasts and liver stellate cells in vitro, and induced the secretion of pro-inflammatory genes. This activation depended on EV-derived which activated in fibroblasts the TNFAIP3-NF- κ B signaling axis. Moreover, they show that CRC cells entrained with activated fibroblasts CM had a growth advantage when injected systemically to immune deficient mice.

The study is interesting, and most of the data are of good quality. Moreover, the human data support the findings and strongly suggest a role for tumor-derived ITGBL1 in progression and survival of colorectal cancer patients. While conceptually not novel- activation of cells in the microenvironment by tumor-derived EVs that forms a hospitable niche- was previously shown, this study adds another mechanistic layer and elucidates a signaling pathway functional in colorectal cancer.

The main concern about the manuscript is the gap between the overstatements and the actual data. This begins in the abstract, but continues throughout the manuscript. For example: "The activated cancer-associated fibroblasts (CAFs) in distal organs

promoted metastatic cancer growth through the TNFAIP3-NF- κ B signaling axis". In fact, this was not shown, but rather that systemic treatment with CRC-derived EVs provided a growth advantage to injected tumor cells. It was not shown that CAFs promoted tumor growth.

Another example from the abstract: "the primary CRC-derived exosomal ITGBL1 stimulated the TNFAIP3-mediated NF- κ B signaling pathway to activate CAFs". In fact, these are not CAFs, but rather normal fibroblasts that were activated. See also comment 14 below for additional concerns regarding overstatements.

It is crucial that the authors make sure that their statements are accurate and backed up by data throughout the manuscript before it can be further considered for publication. The difference between conclusion and speculation is not merely semantic.

Specific comments:

1. The introduction states repeatedly how very little is known about the metastatic niche and how a hospitable metastatic microenvironment is instigated, and about the role of inflammatory cells at

the metastatic site, while ignoring a large body of literature that had already been published. A few examples include:

Various manuscripts by Malanchi I et al., Oskarson T. Nature Med. 2011, Coffelt S. Nature 2015, and many more.

2. In general- the Figure legends are not informative enough. Abbreviations used throughout the figures are not explained anywhere. For example: presumably NC stands for non-coding? This is not explained anywhere, and in Table S5 the control vectors are called non-targeting. So which is it? In addition, the legends should clearly state for the experiments how many times they were performed, how many mice (e.g. in Figure 3), p values, method of quantification, statistical methods used etc.

3. The manuscript text in general is not informative enough regarding what was actually done and sends the reader repeatedly to search for clarifications in the legends and methods. For example: "The purified ITGBL1-rich exosomes promoted the growth of metastatic tumors, whereas silencing ITGBL1 markedly promoted or decreased the growth of metastatic tumors (Fig. 3E, 3F, 3G, 3H)." A search in the legend of this figure did not provide further information. Only a thorough search in the methods revealed that "growth of metastatic tumors" actually means experimental metastasis following IV injection of tumor cells. This is misleading. The text of the manuscript should clearly state what was done, and clarify that the lesions are experimental metastasis.

4. Sequential centrifugation is not enough to definitively define exosomes.

Ultracentrifugation-based exosome preparations contain a mixture of other types of extracellular vesicle falling in the similar size range of exosomes, like apoptotic bodies and membrane particles. The minimal experimental requirements for definition of extracellular vesicles and their functions are detailed in: <https://www.ncbi.nlm.nih.gov/pmc/articles/PMC4275645/>). Therefore, the authors should use the term extracellular vesicles, unless they further characterize the isolated vesicles.

5. Figure 1B, C: what is the difference between CRC primary I and Primary II?

6. Figure 2D-F: the quantification graphs are not clearly linked with the respective blots. Please make it more obvious, for example by labeling them separately.

7. The authors state that: "the exosome-educated fibroblasts or stellate cells from highly metastatic CRC cells expressed higher level of proinflammatory genes, including IL-6, IL-8, IL-1 β , α -SMA, TGF- β , and CXCL12". However, α SMA and TGF β are not pro-inflammatory genes. In fact, they are typical of

myofibroblastic CAF activity, which in many cases is performed by a different subpopulation of CAFs (See Ohlund D. et al. JEM 2017). So in fact, their experiments show that exosomes activate both pro-inflammatory and wound-healing functions in fibroblasts.

8. Figure 3: In Figure 3A, the basal levels of expression with a non-coding control are very low (as expected), and they are upregulated by exosomes containing ITGBL1. However, in Figure 3B (with the same target cells), the control vector (shRNA-NC) induces basal level of expression that are comparable to the ITGBL1 induced gene expression. This is concerning (assuming that NC is indeed the control, which was not clearly stated in the figure legends) and should be explained. Moreover, the in vivo data also suggest that shRNA-NC has an activating and pro-tumorigenic effect of its own. Same is true for Figure 4 Luciferase assays.

9. Figure 4: Western blots should be quantified.

10. Figure 6H: Which of the differences are significant? This should be clearly indicated.

11. Figure S9C-F: The images are not convincing. The co-localizations shown in Figure S9C, E, actually look like they are around alveoli rather than fibroblasts. The co-staining with fibronectin is also not convincing. Can the authors provide higher magnification images to better show cellular co-staining?

12. Figure 7G,H: This assay measured the outgrowth of locally injected colorectal cells in lungs or liver. This is by no means metastasis. The authors should either perform an actual metastatic assay, or use more accurate terms. Referring to these lesions as metastases is misleading.

13. The use of immune deficient mice is a limitation of the study, that should be clearly discussed.

14. In order to establish the mechanism that they propose (in Figure 8), namely, that reprogramming of fibroblasts contributes to the formation of a metastatic niche, the authors should isolate fibroblasts from lungs/liver of mice treated with tumor-derived exosomes and show that they upregulated pro-inflammatory factors. Otherwise, what they actually show is that exosomes can activate fibroblasts in vitro, and that tumor cells pre-treated with fibroblast-secreted factor have a growth advantage in vivo. Overstatements such as "In our study, we found that tumor-derived exosomal ITGBL1 could be directly transferred to lung and liver metastatic niche" should be avoided, or backed up by data.

Reviewer #3 (Remarks to the Author):

In this extensive manuscript Ji and colleagues have investigated the role of ITGBL1 rich exosomes in the activation of fibroblasts into cancer-associated fibroblasts, thereby generating a pre-metastatic niche. They provide a wide range of in vitro and in vivo experiments, providing detailed analysis of potential molecular mechanisms underlying the pro-metastatic effects of these exosomes and support their data with finding in human colorectal cancer specimens. Although some of the molecular mechanisms regarding ITGBL1 regulation (although not in relation to exosomes) have been reported in literature before, the extensive data and thorough analyses provide new evidence for a role of tumor derived exosomes to generate a pre-metastatic niche facilitating metastasis. Although the study is well performed there are a few things which should be addressed to enhance the manuscript/further support the conclusions drawn in the manuscript.

1. The link to immune regulation is somewhat difficult claim since the experiments are all performed in immunodeficient mice. This limits the conclusion with regard to a potential role for the immune system and immunoregulatory cytokines. In the discussion with regard to IL-6 it might be good to include a recent study on the role of IL-6 and hepatocytes in the formation of liver metastases (Lee et al Nature 2019).
2. As the authors indicate in the discussion ITGBL1 promotes breast cancer metastasis via activation of TGF β signalling. Have the authors investigated how TGF β signalling is influenced in the CAFs in vitro and in vivo upon exosomes treatment? This is particularly interesting in the light of the increased TGF β mRNA expression by exosomes and previously published work.
3. Systemic administration of exosomes seems to lead to specific lung and liver metastasis formation. Do the authors have any indication how the exosomes specifically activate fibroblasts in these organs? Or are addition location with metastases/accumulation of exosomes observed? And do the exosomes only accumulate in the CAFs or also in other cell types?
4. Is exosome ITGBL1 related to tissue expression of ITGBL1 in human CRC samples? In addition in discussing the data, if ITGBL1 is linked to distant metastasis it is not surprising that it is linked to overall survival, because of the very well established prognostic impact of having metastasis. A multivariate analysis could reveal if ITGBL1 level is an independent prognostic parameter, or an indicator of metastasis.
5. Does knockdown of RUNX2 affect the total number of exosomes?
6. Overexpression and knockdown experiments have been performed in different cell lines. This makes it somewhat difficult to compare. Based on western blot expression in SW480 for example fig 4A, next to the liver expression knockdown in these cells could provide additional valuable evidence.

7. Figure S10 A/B: staining for aSMA seems not specific. Although clearly CAFs are staining there is a lot of background in surrounding tissue, making the quantification not reliable.

Minor comments:

1. Page 6 line 24, TMN should be TNM
2. Figure 8 could use an upgrade

Manuscript No: NCOMMS-19-14628

We thank the reviewers for the constructive comments, which are very helpful for improving the quality of our work. We have found that the reviewers' comments are very important to strengthen our conclusions. For this reason, we have performed a numerous new experiments to address the reviewers' concerns. It has taken us a relative long time to complete these new experiments. We apologize for the delayed submission of our revised manuscript. Thank you for your understanding. The revised manuscript is in Nature Communications conformation with supplementary information. We believe that the revised manuscript is significantly improved and the manuscript is acceptable for publication in your precious journal.

Our point-by-point responses to the reviewers' comments are as follows:

Reviewers' comments:

Reviewer #1 (Remarks to the Author):

Comment: In the manuscript by Qing Ji et al., the authors demonstrate that Colorectal cancer exosomes expressing Integrin Beta-like 1 activate cancer-associated fibroblasts to promote pro-inflammatory cytokines and EMT programs. The work is timely and extends our knowledge of the importance of tumor exosome proteins in the promotion of metastasis.

Response: We thank the reviewer for considering our being interesting, timely and important. These are very encouraging comments.

Comment: 1. First, the authors need to include important controls, such as normal colon exosomes, in their studies throughout (for example Figure 1B).

Response: These are excellent suggestions. Based on the reviewers' recommendation, we have carefully checked each experiment and included appropriate controls in main figures and supplementary figures. These include (but not limited to): normal plasma EVs from healthy volunteers and EVs prepared from normal colonic epithelial cells. These control results are included in their relative figures of the revised manuscript.

Comment: 2. The authors should not assume that CAFs from the lung and stellate cells from the liver are the primary cells uptaken tumor exosomes. For instance other studies in the liver show that Kupffer cells are the primary cells take up exosomes. The authors need to perform a biodistribution study to show that these cells are the critical players. Moreover, the authors use WI-38 cell in their studies, this cell is a fibroblast and are not CAFs. Thus, the authors need to isolate CAFs from primary and metastatic tumors and incorporate these cells in their analyses.

Response: We thank the reviewer for this valuable comment. We have performed new experiments to study biodistribution of EVs and define the cell types for taking up EVs. For this purpose, EVs were injected into the retro-orbital venous sinus in vivo for 24 h, followed by immunofluorescence analysis. The new results show that CRC-derived EVs were effectively taken up by α -SMA⁺ hStCs (hepatic stellate cells) or fibroblasts, S100A4⁺ fibroblasts, and F4/80⁺ macrophage cells (Kupffer cells). The reviewer is completely correct. The Kupffer cells also effectively take up EVs. By contrast, CD31⁺ endothelial cells in the liver and lung organ are unable to take up EVs, indicating cell specific take-up of EVs. These new results are included in Fig. 3B of the revised manuscript.

In contrast to CRC-derived EVs, EVs isolated from normal colonic epithelial cells NCM-460 were not incorporated by α -SMA⁺ hStCs (hepatic stellate cells) or fibroblasts, S100A4⁺ fibroblasts, and F4/80⁺ macrophage cells, demonstrating that specific targeting by tumor cells-derived EVs. These data are also included in Fig. 3B of the revised manuscript. Together, our data support the fact that α -SMA⁺ hStCs or fibroblasts, S100A4⁺ fibroblasts and F4/80⁺ macrophage cells were the critical players for promoting the initial steps of pre-metastatic niche formation in the liver and lung organ.

Indeed, WI-38 is a non-CAF fibroblast cell line. Using above lung and liver metastatic tissues from CRC patients, we showed a high density of α -SMA⁺ fibroblasts around the metastatic tumors (*SI Appendix*, Fig. S16). To strengthen our conclusions, we isolated primary CAFs from the liver metastatic tumors of CRC patients and primary NFs isolated from normal liver tissues as a control. Expression levels of ITGBL1 in the primary NFs and CAFs from several different CRC patients were analyzed. CAFs with low-ITGBL1 and high-ITGBL1 expression were used for the subsequent experiments. These new data were included in *SI Appendix*, Fig. S17A, S17B. Expectedly, primary CAFs with high ITGBL1 expression secreted high levels of IL-6 and IL-8 (*SI Appendix*, Fig. S17C, S17D). Similar to fibroblasts educated by ITGBL1-rich EVs, primary CAFs with high ITGBL1 markedly contributed to tumorigenicity (*SI Appendix*, Fig. S17E, S17F), tumor stemness (*SI Appendix*, Fig. S17G), tumor metastatic growth (*SI Appendix*, Fig. S17H, S17I), and tumor EMT (*SI Appendix*, Fig. S17J). These new data were included in the revised version of the manuscript.

In vitro studies also validated the fact that Primary CAFs with high ITGBL1 expression promoted the stemness genes expression, spheroid formation ability (*SI Appendix*, Fig. S18A, S18B), the positive regulatory effect on EMT-associated gene expression, tumor cell migration (*SI Appendix*, Fig. S18C, S18D).

Independent evidence was obtained from lung and liver fibroblasts of tumor-derived EVs-treated mice and the results showed up-regulation of pro-inflammatory factors in ITGBL1-rich EVs-treated fibroblasts or stellate cells (Fig. 3C, 3D), relative to normal control NCM-460-EVs education. Moreover, education of ITGBL1-enriched EVs from CRC cells increased the frequency of α -SMA⁺ hStCs or fibroblasts, FN deposition, F4/80⁺ macrophage, and Ly6G⁺ myeloid cell migration

(Fig. 3E, 3F). These findings suggest that CRC-ITGBL1-enriched EVs contribute to pre-metastatic niche formation in the liver and lung. Most importantly, we isolated CAFs from the metastatic tumors in the lung and liver treated with tumor-derived EVs, and the results showed the increased mRNA expression of IL-6, IL-8, IL-1 β , α -SMA, TGF- β , and CXCL12 in ITGBL1-enriched EVs-treated CAFs relative to normal EVs. Together, these independent lines of evidence support our conclusions that ITGBL1⁺ CAFs significantly contributed to cancer metastasis.

Comment: 3. It is also not clear how integrin beta-like 1 compared to B1 integrin. To demonstrate beta-like 1 intern's role, the authors should also knockdown B1 integrin. Does B-like 1 integrin bind to other integrins? what ligands does it bind to, other ECM molecules? The paper by A. Hoshino et al also demonstrated that a6b1 integrin is relevant for lung mets.

Response: We thank the reviewer for this constructive comment. Using co-immunoprecipitation (Co-IP) and immunoblot analysis, our previous studies showed that TNFAIP3 as a new binding partner of ITGBL1 in WI-38 or LX-2 cells. Co-IP in combination with LC-MS/MS experiments shows no significant and direct binding between ITGBL1 and other integrins (*SI Appendix*, Data S1). Co-IP in combination with immunoblot experiments further validates these findings (*SI Appendix*, Fig. S13A, S13B). Importantly, silencing integrin β 1 barely affected the regulatory effect of ITGBL1 on the NF- κ B signaling pathway (*SI Appendix*, Fig. S13C, S13D). These findings demonstrate that ITGBL1 has no direct interaction with B1 integrin. We have included these new data in the revised manuscript.

Comment: 4. Do the authors see a premetastatic site with exosome injection.

Response: We agree. Systemic injection of EVs via the retro-orbital venous sinus in vivo for 3 weeks followed by immunofluorescence analysis showed that treatment with ITGBL1-enriched EVs prepared from CRCs increased the α -SMA⁺ hStCs and myofibroblasts, FN deposition, F4/80⁺ macrophages, and migration of Ly6G⁺ myeloid cells in the liver or lung (Fig. 3C, 3D). As a control, NCM-460-EVs did not induce this phenotypic change. These results demonstrated that ITGBL1-enriched EVs established a pre-metastatic niche formation in the liver and lung. Thus, CRC cells-derived EVs are involved in a premetastatic niche formation. We have included these new data in the revised version of the manuscript.

Comment: 5. These models are not necessarily organotropic but clearly are promoting metastasis to lung and liver. Can they promote metastasis to bone for instance as well or is it specific to lung and liver.

Response: Agree. In an orthotopic CRC model in which tumor cells were implanted in the cecum, we only observed metastasis in the liver and lung (a small amount). Other sites including brain and bone lacked detectable metastases (Fig. S6A-S6B).

These findings show that metastasis may be specific to liver and lung. These data are included in Fig. 4A-4F of the revised manuscript.

Comment: 6. Can the authors provide histology in all metastatic tissues.

Response: Agree. In the revised manuscript, we have provided histological studies in all metastasis related studies.

Comment: 7. Details are often missing, as in Figure 1F, what is each sample specifically.

Response: We apologize for not providing details in the original version of the manuscript. In the revised manuscript, we provide detailed information of results and experimental procedures. We have carefully read the manuscript and detailed information was provided in the main text, figures, figure legends, and methods.

Comment: 8. Interpretation of data is often not visualized in the panels, as with the spheroid numbers seem not to alter in Anti-IL-6 and anti-IL-8 experiments in the panels.

Response: Agree. We have repeated the spheroid formation experiments and provided the new data in the figures. We have carefully checked the manuscript to ensure that our interpretation and claims were supported by panel figures.

Comment: 9. Mechanistically, the connections are missing for instance how is Runx2 related to EMT here.

Response: Agree. We apologize for not providing the mechanistic information between Runx2 and EMT. In the revised manuscript, we have provided the mechanistic information of the Runx2-ITGBL1-EMT axis (Fig. 2, Fig. 8, and new *SI Appendix*, Fig. S19). This information is now included in the revised manuscript.

Reviewer #2 (Remarks to the Author):

Comment: In this study, Ji et al. show that colorectal cancer cell lines secrete extracellular vesicles (EVs) containing high levels of integrin beta-like 1 (ITGBL1). Furthermore, these findings are nicely validated in human data from CRC patients and correlated with metastatic disease and with survival. CRC-derived EVs activated human lung fibroblasts and liver stellate cells in vitro, and induced the secretion of pro-inflammatory genes. This activation depended on EV-derived which activated in fibroblasts the TNFAIP3-NF- κ B signaling axis. Moreover, they show that CRC cells

entrained with activated fibroblasts CM had a growth advantage when injected systemically to immune deficient mice.

The study is interesting, and most of the data are of good quality. Moreover, the human data support the findings and strongly suggest a role for tumor-derived ITGBL1 in progression and survival of colorectal cancer patients. While conceptually not novel- activation of cells in the microenvironment by tumor-derived EVs that forms a hospitable niche- was previously shown, this study adds another mechanistic layer and elucidates a signaling pathway functional in colorectal cancer.

Response: We thank the reviewer for considering our work being interesting and our data in good quality. We particularly appreciate that the reviewer considering the clinical data being important. Also, we thank the reviewer for considering our findings being mechanistically informative.

Comment: The main concern about the manuscript is the gap between the overstatements and the actual data. This begins in the abstract, but continues throughout the manuscript. For example: “The activated cancer-associated fibroblasts (CAFs) in distal organs promoted metastatic cancer growth through the TNFAIP3-NF- κ B signaling axis”. In fact, this was not shown, but rather that systemic treatment with CRC-derived EVs provided a growth advantage to injected tumor cells. It was not shown that CAFs promoted tumor growth.

Response: Completely agree. We apologize for this overstatement without supportive experimental data. In the revised manuscript, we have revised this statement to accurately justify our findings and conclusions.

Comment: Another example from the abstract: “the primary CRC-derived exosomal ITGBL1 stimulated the TNFAIP3-mediated NF- κ B signaling pathway to activate CAFs”. In fact, these are not CAFs, but rather normal fibroblasts that were activated. See also comment 14 below for additional concerns regarding overstatements. It is crucial that the authors make sure that their statements are accurate and backed up by data throughout the manuscript before it can be further considered for publication. The difference between conclusion and speculation is not merely semantic.

Response: Completely agree. We tremendously appreciate the reviewer’s suggestion. We apologize for including the speculative statement in the abstract. In the revised manuscript, we have corrected this sentence and indeed the activated normal fibroblasts rather than CAFs were used. In the revised manuscript, we have carefully read the manuscript to sure that our statements are backed up by experimental data. For this reason, we have performed many new experiments to further support and strengthen our conclusions. These new experimental data are now included in their respective sections of the revised manuscript.

Specific comments:

Comment: 1. The introduction states repeatedly how very little is known about the metastatic niche and how a hospitable metastatic microenvironment is instigated, and about the role of inflammatory cells at the metastatic site, while ignoring a large body of literature that had already been published. A few examples include: Various manuscripts by Malanchi I et al., Oskarsson T. *Nature Med.* 2011, Coffelt S. *Nature* 2015, and many more.

Response: Completely agree. We apologize for ignoring the published important literatures related to metastatic niches. Indeed, an increasingly accumulative data support the concept of a hospitable metastatic microenvironment in supporting cancer metastasis. In the revised manuscript, we have cited the relevant manuscript, including:

- [1] Malanchi I, et al. (2011) Interactions between cancer stem cells and their niche govern metastatic colonization. *Nature* 481(7379):85-89.
- [2] Oskarsson T, et al. (2011) Breast cancer cells produce tenascin C as a metastatic niche component to colonize the lungs. *Nat Med* 17(7):867-874.
- [3] Costa-Silva B, et al. (2015) Pancreatic cancer exosomes initiate pre-metastatic niche formation in the liver. *Nat Cell Biol* 17(6):816-826.
- [4] Hara T, et al. (2017) Control of metastatic niche formation by targeting APBA3/Mint3 in inflammatory monocytes. *Proc Natl Acad Sci U S A* 114(22):E4416-E4424.
- [5] Wculek SK, Malanchi I. (2015) Neutrophils support lung colonization of metastasis-initiating breast cancer cells. *Nature* 528(7582):413-417.
- [6] Coffelt SB, et al. (2015) IL-17-producing $\gamma\delta$ T cells and neutrophils conspire to promote breast cancer metastasis. *Nature* 522(7556):345-348.
- [7] Nielsen SR, et al. (2016) Macrophage-secreted granulins supports pancreatic cancer metastasis by inducing liver fibrosis. *Nat Cell Biol* 18(5):549-560.
- [8] Lee JW, et al. (2019) Hepatocytes direct the formation of a pro-metastatic niche in the liver. *Nature* 567(7747):249-252.

Comment: 2. In general- the Figure legends are not informative enough. Abbreviations used throughout the figures are not explained anywhere. For example: presumably NC stands for non-coding? This is not explained anywhere, and in Table S5 the control vectors are called non-targeting. So which is it? In addition, the legends should clearly state for the experiments how many times they were performed, how many mice (e.g. in Figure 3), p values, method of quantification, statistical methods used etc.

Response: These are excellent comments. We apologize for not providing detailed information in the text and figure legends. In the revised manuscript, we have carefully read through the manuscript to make sure that all abbreviations are

explained and spelt out throughout the entire manuscript. In the figure legend section, we produced the mouse, tissue, and other sample numbers, experimental times, quantification methods, and statistical values. We thank the reviewer once again for this important suggestion.

Comment: 3. The manuscript text in general is not informative enough regarding what was actually done and sends the reader repeatedly to search for clarifications in the legends and methods. For example: “The purified ITGBL1-rich exosomes promoted the growth of metastatic tumors, whereas silencing ITGBL1 markedly promoted or decreased the growth of metastatic tumors (Fig. 3E, 3F, 3G, 3H).” A search in the legend of this figure did not provide further information. Only a thorough search in the methods revealed that “growth of metastatic tumors” actually means experimental metastasis following IV injection of tumor cells. This is misleading. The text of the manuscript should clearly state what was done, and clarify that the lesions are experimental metastasis.

Response: We thank the reviewer for these constructive comments, which are very important to improve the quality of our work. Also, we apologize for poor clarification of our findings and any confusion for readers. In the revised manuscript, we have carefully read through the manuscript to make sure that the text is clearly written. These include the main text, supplementary information, methods, and figure legends. In particular, the original data presented in Fig. 3E, 3F, 3G, and 3H were clearly presented “the metastatic lesions were obtained from an experimental lung metastasis model by lateral tail vein injection and from a liver metastasis model by intrasplenic injection models in nude mice” in the revised manuscript.

Comment: 4. Sequential centrifugation is not enough to definitively define exosomes. Ultracentrifugation-based exosome preparations contain a mixture of other types of extracellular vesicle falling in the similar size range of exosomes, like apoptotic bodies and membrane particles. The minimal experimental requirements for definition of extracellular vesicles and their functions are detailed in: <https://www.ncbi.nlm.nih.gov/pmc/articles/PMC4275645/>). Therefore, the authors should use the term extracellular vesicles, unless they further characterize the isolated vesicles.

Response: Completely agree. We thank the reviewer for the excellent supervision. In the revised manuscript, we have replaced exosomes with extracellular vesicles (EVs) throughout the entire manuscript. We have cited the publication:

[1] Lövall J, et al. (2014) Minimal experimental requirements for definition of extracellular vesicles and their functions: a position statement from the International Society for Extracellular Vesicles. *J Extracell Vesicles* 3:26913.

Comment: 5. Figure 1B, C: what is the difference between CRC primary I and Primary II?

Response: Primary CRC I is a primary CRC tumor without paired metastatic tissues, and primary CRC II is the primary CRC tumors with paired metastatic tissues (Met II). In the revised legends of Figure 1B, C, we have provided detailed interpretation for CRC primary I and Primary II.

Comment: 6. Figure 2D-F: the quantification graphs are not clearly linked with the respective blots. Please make it more obvious, for example by labeling them separately.

Response: In the revised manuscript, we have separated the Western blot figures with the respective quantification graphs to make the data representation clearly.

Comment: 7. The authors state that: “the exosome-educated fibroblasts or stellate cells from highly metastatic CRC cells expressed higher level of proinflammatory genes, including IL-6, IL-8, IL-1 β , α -SMA, TGF- β , and CXCL12”. However, α SMA and TGF β are not pro-inflammatory genes. In fact, they are typical of myofibroblastic CAF activity, which in many cases is performed by a different subpopulation of CAFs (See Ohlund D. et al. JEM 2017). So in fact, their experiments show that exosomes activate both pro-inflammatory and wound-healing functions in fibroblasts.

Response: Completely agree. Indeed, the α SMA- and TGF β -positive cells represent a typical myofibroblastic population of CAFs, which are rather invasive. The reviewer is completely correct about interpretation of our data that exosomes activate both pro-inflammatory and wound-healing functions in fibroblasts. In the revised manuscript, we clearly stated “the EVs-educated fibroblasts or stellate cells from highly metastatic CRC cells expressed higher level of proinflammatory genes, typical marker genes of myofibroblastic CAF activity, and chemokines, including IL-6, IL-8, IL-1 β , α -SMA, TGF- β and CXCL12”. In the Discussion, we have cited the corresponding reference and included the discussion of this important issue.

[1] Öhlund D, et al. (2017) Distinct populations of inflammatory fibroblasts and myofibroblasts in pancreatic cancer. *J Exp Med* 214(3):579-596.

Comment: 8. Figure 3: In Figure 3A, the basal levels of expression with a non-coding control are very low (as expected), and they are upregulated by exosomes containing ITGBL1. However, in Figure 3B (with the same target cells), the control vector (shRNA-NC) induces basal level of expression that are comparable to the ITGBL1 induced gene expression. This is concerning (assuming that NC is indeed the control, which was not clearly stated in the figure legends) and should be explained. Moreover, the in vivo data also suggest that shRNA-NC has an activating and pro-tumorigenic effect of its own. Same is true for Figure 4 Luciferase assays.

Response: We thank the reviewer for this important suggestion. In the original data

presented Fig. S5, we showed that the ITGBL1 level in SW620 EVs was higher than that in SW480 EVs. We also showed that in new Fig. S9 the level of cytokines was higher in SW620 EVs-educated WI-38 and LX-2 than SW480 EVs-educated WI-38 and LX-2. These results explained why the control SW620 EVs(shRNA-NC)-induced basal expression levels were comparable to the ITGBL1-overexpressing SW480 EVs (pcDNA3.1-ITGBL1)-induced gene expression. In summary, the basal levels of gene expression in SW480 EVs- and SW620 EVs-treated WI-38/LX-2 cells are different (Fig. S9). For better understanding, we investigated gene expression levels in the WI-38/LX-2 cells received the same treatment condition, including overexpression and knockdown. These include normal EVs- and control non-treated samples. These new experimental data were included in the revised manuscript (Fig. S10).

In the in vivo setting, luciferase-labeled HCT116 cells were used for experimental lung and liver metastasis by tail vein injection and intrasplenic injection (new Fig. 4A-4D). Without EVs treatment, the luciferase-labeled HCT116 cells formed distal metastatic lesions in lung and liver. The CRC EVs promoted the growth of distal metastases, but the normal EVs lacked such a capacity. Therefore, in compared with the control non-treated HCT116 group, CRC-SW620 (shRNA-NC) EVs promoted the metastatic tumor growth. Similar results were also observed in the orthotopic CRC metastasis model (new Fig. 4E-4F).

In the original data presented Figure 4, luciferase assay showed that the similar as above detection of gene expression (IL-6, IL-8, IL-1 β , α -SMA, TGF- β , and CXCL12). For better understanding, we investigated the luciferase activity in WI-38 and LX-2 cells under the same treatment condition, including overexpression and knockdown (new Fig. 5G-5H, Fig. S11C-11D). These new experimental data are now included in the revised manuscript.

Comment: 9. Figure 4: Western blots should be quantified.

Response: Agree. In the revised manuscript, we have provided quantitative data of Western blot in all analyses and all figures throughout the manuscript.

Comment: 10. Figure 6H: Which of the differences are significant? This should be clearly indicated.

Response: Agree. In the revised version of manuscript, we have provided indicative markers to show significant differences in Figure 6H (new Fig. 7J).

Comment: 11. Figure S9C-F: The images are not convincing. The co-localizations shown in Figure S9C, E, actually look like they are around alveoli rather than fibroblasts. The co-staining with fibronectin is also not convincing. Can the authors provide higher magnification images to better show cellular co-staining?

Response: We thank the reviewer for this constructive comment. We have performed a biodistribution study to define cell types that take up tumor-derived EVs. Injection of EVs via the retro-orbital venous sinus *in vivo* for 24 h followed by immunofluorescence analysis showed that CRC-derived EVs were efficiently incorporated by α -SMA⁺ hStCs (hepatic stellate cells), myofibroblasts, S100A4⁺ fibroblasts, and F4/80⁺ macrophage cells. These EVs were unable to be taken up by CD31⁺ endothelial cells in the liver and lung (Fig. 3B). In contrast, the control EVs isolated from normal colonic epithelial cells NCM-460 were not efficiently incorporated by above cells (Fig. 3B). These data suggest that α -SMA⁺ hStCs, myofibroblasts, S100A4⁺ fibroblasts, and F4/80⁺ macrophage cells were the critical players for promoting the initial steps of pre-metastatic niche formation in the liver and lung.

Comment: 12. Figure 7G, H: This assay measured the outgrowth of locally injected colorectal cells in lungs or liver. This is by no means metastasis. The authors should either perform an actual metastatic assay, or use more accurate terms. Referring to these lesions as metastases is misleading.

Response: We thank the reviewer for raising this important issue. We agree with the reviewer's suggestion. Indeed, the experimental metastasis model is not a clinically relevant metastasis model. To experimentally address the reviewer's concern, we have performed a clinically relevant orthotopic model by implanting luciferase-labeled HCT116 cells in the cecum. Our results were further validated by *in vivo* imaging. The conditioned medium from ITGBL1-overexpressing WI-38 or LX-2 promoted the growth of metastatic tumors in the liver (new Figure 8G, 8H). These new metastatic data were included in the revised manuscript.

Comment: 13. The use of immune deficient mice is a limitation of the study, that should be clearly discussed.

Response: Agree. In the discussion section, we have discussed the possible limitations of immunodeficient mice in our model system: "Although we have shown the evidence in the immune deficient mice that ITGBL1-rich EVs derived from highly metastatic cancer cells accelerate metastatic cancer growth through a CAF activation mechanism, the using model of the athymic nude mice has certain limitations. Deficiency of T-lymphocytes considerably immunocompromises the mice and enables the engraftment, growth and eventually metastasizing of the tumor cells from the subcutaneous or orthotopic tumor xenograft (53). However, the athymic nude mice lack the proper T-lymphocytes immune response at the site of primary tumor (54), which could not completely reflect the clinical scenario of primary tumor (55). Therefore, considering the effect of host immune system, we explored the corresponding biological function and mechanism in immunocompetent C57Bl/6 mice."

In order to experimentally address this issue, we have performed a study in

immunocompetent C57Bl/6 mice. In consistent with the results seen in immunodeficient nude mice, CRC-ITGBL1-enriched EVs increased the frequency of α -SMA⁺ hStCs, myofibroblasts, FN deposition, F4/80⁺ macrophage, and Ly6G⁺ myeloid cell migration to the liver or lung of C57Bl/6 mice (*SI Appendix*, Fig. S21A, S21B), supporting the fact of a pre-metastatic niche formation. Importantly, both experimental metastasis model and the cecum orthotopic metastasis model showed that ITGBL1-rich EVs accelerated the growth of metastatic tumors (*SI Appendix*, Fig. S22A, S22B, S22C), and the increased mRNA expression of IL-6, IL-8, IL-1 β , α -SMA, TGF- β , and CXCL12 (*SI Appendix*, Fig. S22D, S22E, S22F). These findings were also observed in ITGBL1-rich EVs-treated CAFs. All these results were performed in C57Bl/6 mice and supported the data obtained from the immunodeficient nude mice. However, due to the limitations of existence of autoimmune regulation in C57Bl/6 mice, the regulatory effect of ITGBL1-enriched EVs on EVs incorporation, EV fusion with recipient cells, the pre-metastatic niche formation, and tumor metastatic growth warrant thorough investigation in the future. Nevertheless, these preliminary results from the C57Bl/6 competent mice support the general conclusion of our findings.

Comment: 14. In order to establish the mechanism that they propose (in Figure 8), namely, that reprogramming of fibroblasts contributes to the formation of a metastatic niche, the authors should isolate fibroblasts from lungs/liver of mice treated with tumor-derived exosomes and show that they upregulated pro-inflammatory factors. Otherwise, what they actually show is that exosomes can activate fibroblasts in vitro, and that tumor cells pre-treated with fibroblast-secreted factor have a growth advantage in vivo. Overstatements such as “In our study, we found that tumor-derived exosomal ITGBL1 could be directly transferred to lung and liver metastatic niche” should be avoided, or backed up by data.

Response: Completely agree. We thank the reviewer for this excellent comment. To strengthen our conclusions proposed in Figure 8 of the original manuscript (now Figure 9 of the revised manuscript), we have experimentally addressed the reviewer’s comment. We isolated lung and liver fibroblasts from mice receiving treatment with tumor-derived EVs for three weeks. Our results showed up-regulation of pro-inflammatory factors in ITGBL1-enriched EVs-treated fibroblasts and stellate cells (Fig. 3C, 3D). In contrast, normal control NCM-460-EVs education for three weeks did not produce this effect. Moreover, education of CRC-ITGBL1-enriched EVs increased the frequency of α -SMA⁺ hStCs, myofibroblasts, FN deposition, F4/80⁺ macrophage, and Ly6G⁺ myeloid cell migration to the liver (Fig. 3E, *SI Appendix*, Fig. S5B, S5C, S5D, S5E) or lung (Fig. 3F, *SI Appendix*, Fig. S5F, S5G, S5H, S5I). These data demonstrated that CRC-ITGBL1-enriched EVs promoted pre-metastatic niche formation in the liver and lung. Importantly, we isolated cancer associated fibroblasts (CAFs) from the metastatic tumors in the lung and liver receiving treatment with tumor-derived EVs. Our results showed the increased mRNA expression of IL-6, IL-8, IL-1 β , α -SMA, TGF- β , and CXCL12 in ITGBL1-enriched

EVs-treated CAFs relative to normal EVs (Fig. 4H, 4I, 4J).

In the revised manuscript, we have deleted the sentence “In our study, we found that tumor-derived exosomal ITGBL1 could be directly transferred to lung and liver metastatic niche”.

Reviewer #3 (Remarks to the Author):

Comment: In this extensive manuscript Ji and colleagues have investigated the role of ITGBL-1 rich exosomes in the activation of fibroblasts into cancer-associated fibroblasts, thereby generating a pre-metastatic niche. They provide a wide range of in vitro and in vivo experiments, providing detailed analysis of potential molecular mechanisms underlying the pro-metastatic effects of these exosomes and support their data with finding in human colorectal cancer specimens. Although some of the molecular mechanisms regarding ITGBL1 regulation (although not in relation to exosomes) have been reported in literature before, the extensive data and thorough analyses provide new evidence for a role of tumor derived exosomes to generate a pre-metastatic niche facilitating metastasis. Although the study is well performed there are a few things which should be addressed to enhance the manuscript/further support the conclusions drawn in the manuscript.

Response: We thank the reviewer for these positive comments, which are constructive and encouraging for us to pursue future research along this line. We tremendously appreciate these constructive comments.

Comment: 1. The link to immune regulation is somewhat difficult claim since the experiments are all performed in immunodeficient mice. This limits the conclusion with regard to a potential role for the immune system and immunoregulatory cytokines. In the discussion with regard to IL-6 it might be good to include a recent study on the role of IL-6 and hepatocytes in the formation of liver metastases (Lee et al Nature 2019).

Response: We thank the reviewer for this excellent comment. In the discussion section, we have discussed the possible limitations of immunodeficient mice in our model system: “Although we have shown the evidence in the immune deficient mice that ITGBL1-rich EVs derived from highly metastatic cancer cells accelerate metastatic cancer growth through a CAF activation mechanism, the using model of the athymic nude mice has certain limitations. Deficiency of T-lymphocytes considerably immunocompromises the mice and enables the engraftment, growth and eventually metastasizing of the tumor cells from the subcutaneous or orthotopic tumor xenograft (53). However, the athymic nude mice lack the proper T-lymphocytes immune response at the site of primary tumor (54), which could not completely

reflect the clinical scenario of primary tumor (55). Therefore, considering the effect of host immune system, we explored the corresponding biological function and mechanism in immunocompetent C57Bl/6 mice.”

In order to experimentally address this issue, we have performed a study in immunocompetent C57Bl/6 mice. In consistent with the results seen in immunodeficient nude mice, CRC-ITGBL1-enriched EVs increased the frequency of α -SMA⁺ hStCs, myofibroblasts, FN deposition, F4/80⁺ macrophage, and Ly6G⁺ myeloid cell migration to the liver or lung of C57Bl/6 mice (*SI Appendix*, Fig. S21A, S21B), supporting the fact of a pre-metastatic niche formation. Importantly, both experimental metastasis model and the cecum orthotopic metastasis model showed that ITGBL1-rich EVs accelerated the growth of metastatic tumors (*SI Appendix*, Fig. S22A, S22B, S22C), and the increased mRNA expression of IL-6, IL-8, IL-1 β , α -SMA, TGF- β , and CXCL12 (*SI Appendix*, Fig. S22D, S22E, S22F). These findings were also observed in ITGBL1-rich EVs-treated CAFs. All these results were performed in C57Bl/6 mice and supported the data obtained from the immunodeficient nude mice. However, due to the limitations of existence of autoimmune regulation in C57Bl/6 mice, the regulatory effect of ITGBL1-enriched EVs on EVs incorporation, EV fusion with recipient cells, the pre-metastatic niche formation, and tumor metastatic growth warrant thorough investigation in the future. Nevertheless, these preliminary results from the C57Bl/6 competent mice support the general conclusion of our findings.

In the revised Discussion, we have also discussed the important role of IL-6 and hepatocytes in the formation of liver metastases (Lee JW, et al. Nature, 2019). The article by Lee et al., “(Lee JW, et al. (2019) Hepatocytes direct the formation of a pro-metastatic niche in the liver. Nature 567(7747):249-252.)” is now cited in the revised manuscript.

Comment: 2. As the authors indicate in the discussion ITGBL1 promotes breast cancer metastasis via activation of TGF β signaling. Have the authors investigated how TGF β signalling is influenced in the CAFs in vitro and in vivo upon exosomes treatment? This is particularly interesting in the light of the increased TGF β mRNA expression by exosomes and previously published work.

Response: Completely agree. These are excellent suggestions. In the revised manuscript, we have provided quantitative data of TGF- β . TGF- β levels in EVs were not alerted in ITGBL1 overexpressing or silencing CRC cells (*SI Appendix*, Fig. S20A). Accordingly, the effect on TGF- β /Smads signaling pathway was marginal in the targeted HCT116 cells or CAFs from liver metastatic tumors (*SI Appendix*, Fig. S20B, S20C). Nevertheless, we observed increased secretion of TGF- β in the activated LX-2 upon indicated EVs treatment (*SI Appendix*, Fig. S20D), accompanying with positive effect on TGF- β /Smads signaling pathway in HCT116 cells (*SI Appendix*, Fig. S20E, S20F). These new data are now included in the revised version

of the manuscript.

Comment: 3. Systemic administration of exosomes seems to lead to specific lung and liver metastasis formation. Do the authors have any indication how the exosomes specifically activate fibroblasts in these organs? Or are addition location with metastases/accumulation of exosomes observed? And do the exosomes only accumulate in the CAFs or also in other cell types?

Response: Agree. To study tissue distribution of EVs, the purified EVs were systemically injected into mice via the retro-orbital venous sinus according to the protocols provided by Hoshino et al (18). *In vivo* tracing of the injected EVs showed that CRC-derived EVs were accumulated in lung or liver, and very weak signals in the brain and bone marrow (Fig. 3A, *SI Appendix*, Fig. S5A).

We have also performed new experiments to study biodistribution of EVs and define the cell types for taking up EVs. For this purpose, EVs were injected into the retro-orbital venous sinus *in vivo* for 24 h, followed by immunofluorescence analysis. The new results show that CRC-derived EVs were effectively taken up by α -SMA⁺ hStCs (hepatic stellate cells) or fibroblasts, S100A4⁺ fibroblasts, and F4/80⁺ macrophage cells (Kupffer cells). The reviewer is completely correct. The Kupffer cells also effectively take up EVs. By contrast, CD31⁺ endothelial cells in the liver and lung organ are unable to take up EVs, indicating cell specific take-up of EVs. These new results are included in Fig. 3B of the revised manuscript.

In the *in vivo* setting, mice were pretreated by retro-orbital injection for 3 weeks with EVs containing different levels of ITGBL1, following by intravenous injection with tumor cells via the lateral tail vein or spleen to establish an experimental lung and liver metastasis model. In Fig. 4A, *in vivo* imaging data showed that the purified ITGBL1-enriched EVs promoted the growth of lung metastatic tumors, whereas silencing ITGBL1 markedly decreased the growth of lung metastatic tumors. H&E staining of lung metastatic tissue sections further validated the tumor promoting effect of ITGBL1-enriched EVs and silencing ITGBL1 largely reduced the lung metastatic foci (Fig. 4B). In an experimental liver metastasis model, the representative *in vivo* imaging pictures and H&E staining images also demonstrated the promoting effect of ITGBL1-enriched EVs on the growth of liver metastatic tumors (Fig. 4C, Fig. 4D). Importantly, the cecum orthotopic implantation model showed accelerated liver metastatic tumor growth in ITGBL1-enriched EVs-treated mice (Fig 4E, 4F). Nearly all the CRC-EVs treated mice developed liver metastases, whereas only 16.7% to 33.3% of the CRC-EVs treated mice developed lung metastases. However, in the brain and bone tissues of indicated mice, no obvious metastatic foci were detected (*SI Appendix*, Fig. S6A, S6B). α -SMA is a characteristic marker of CAFs. Treatment with ITGBL1-enriched EVs increased the α -SMA⁺ CAFs in metastatic lesions (Fig. 4G).

Comment: 4. Is exosome ITGBL1-related to tissue expression of ITGBL1 in human

CRC samples? In addition in discussing the data, if ITGBL1 is linked to distant metastasis it is not surprising that it is linked to overall survival, because of the very well established prognostic impact of having metastasis. A multivariate analysis could reveal if ITGBL1 level is an independent prognostic parameter, or an indicator of metastasis.

Response: We thank the reviewer for this excellent suggestion. According to the reviewer's suggestion, we have performed the Pearson correlation analysis to correlate the relationship ITGBL1 EVs expression in CRC plasma and ITGBL1 mRNA expression in CRC tissues. Multivariate Cox regression models were used to analyze the correlations between risk factors and survival outcomes. SPSS 22.0 software was used for statistical analyses. The results demonstrated that the mRNA expression of ITGBL1 in human CRC tumor samples had strong correlation with the ITGBL1 levels in corresponding plasma EVs of CRC patients (*SI Appendix*, Fig. S3A). Multivariate Cox regression analysis further revealed that both ITGBL1 levels in EVs and distant metastasis served as independent predictors of poor prognosis in CRC patients (Table S3). We have included these findings in the revised manuscript.

Comment: 5. Does knockdown of RUNX2 affect the total number of exosomes?

Response: We have performed a new experiment to demonstrate that the total numbers of EVs from an equal number of cells were increased or decreased by overexpressing and knocking down RUNX2 (Fig. 2I). Additionally, the total numbers of EVs from an equal number of cells were also up-regulated or down-regulated by nSMase2 overexpression or knockdown, respectively (Fig. 2M). We have included these new experimental data in the revised version of the manuscript.

Comment: 6. Overexpression and knockdown experiments have been performed in different cell lines. This makes it somewhat difficult to compare. Based on western blot expression in SW480 for example fig 4A, next to the liver expression knockdown in these cells could provide additional valuable evidence.

Response: Agree. To improve comparison parameters, we have provided new western blot results to show the protein expression in same CRC cells. Both overexpression and knockdown experiments were performed using the same cell line (new Fig. 5C, 5D, 5E, 5F). These data were included in the revised version of manuscript.

Comment: 7. Figure S10 A/B: staining for α SMA seems not specific. Although clearly CAFs are staining there is a lot of background in surrounding tissue, making the quantification not reliable.

Response: Agree. We have performed a new immunofluorescence staining to detect the α -SMA expression in CAFs (new Fig. 4G), and the quantitative results were presented in new SI Appendix, Fig. S6C, S6D, S6E. These new results demonstrate

the specific staining of α -SMA. These new findings are now included in the revised manuscript.

Minor comments:

Comment: 1. Page 6 line 24, TMN should be TNM

Response: We have replaced “TMN” with “TNM”.

Comment: 2. Figure 8 could use an upgrade

Response: We have tried our best to upgrade the Figure 8 (new Figure 9).

REVIEWERS' COMMENTS:

Reviewer #1 (Remarks to the Author):

The authors have addressed all concerns raised by this reviewer.

The manuscript has improved greatly increasing the novelty of the work. There are no other queries.

Reviewer #2 (Remarks to the Author):

In the revised version the authors have addressed all my comments and the manuscript is much improved.

In particular, addition of in vivo data in immune competent mice and metastases following orthotopic injection of tumor cells contributed to the validity of the conclusions.

One clarification remains: the authors have isolated CAFs from metastases (new Fig. 4H-J), but did not report how isolation was performed. This is important because CAF markers are controversial and different labs refer to different cells as "CAF". This should be clearly stated in the text and/or figure legends.

Following this minor revision, I now recommend publication of this manuscript.

Reviewer #3 (Remarks to the Author):

The authors have addressed my comments carefully by performing many additional experiments.

Manuscript No: NCOMMS-19-14628A

We thank the reviewers for the constructive comments, which are very helpful for improving the quality of our work.

Our point-by-point responses to the reviewers' comments are as follows:

Reviewers' comments:

Reviewer #1 (Remarks to the Author):

Comment: The authors have addressed all concerns raised by this reviewer. The manuscript has improved greatly increasing the novelty of the work. There are no other queries.

Response: Thanks to the reviewer's comment.

Reviewer #2 (Remarks to the Author):

Comment: 1. In the revised version the authors have addressed all my comments and the manuscript is much improved. In particular, addition of in vivo data in immune competent mice and metastases following orthotopic injection of tumor cells contributed to the validity of the conclusions.

Response: Thanks to the reviewer's good comment.

Comment: 2. One clarification remains: the authors have isolated CAFs from metastases (new Fig. 4H-J), but did not report how isolation was performed. This is important because CAF markers are controversial and different labs refer to different cells as "CAF". This should be clearly stated in the text and/or figure legends. Following this minor revision, I now recommend publication of this manuscript.

Response: We thank the reviewer for this valuable comment. We apologize for not providing details that how isolation was performed. In the revised manuscript, we provide detailed information of experimental procedures as follows:

Isolation and primary culture of CAFs

The primary CAFs (cancer-associated fibroblasts) or NFs (normal fibroblasts) were isolated from fresh liver metastatic tumor tissues or normal liver tissues^{1,2}. Briefly, tumor tissues were cut into small pieces of 1 mm³ using a razor blade, and the tissue pieces were dissociated using the Tumor Dissociation Kit (Miltenyi Biotec) according to the manufacturer's procedure. Cells were then resuspended, passed through a cell

strainer (100 μ M), and finally plated into a T75 flask. Tissue blocks trapped in the cell strainer were seeded in 10 cm² culture dishes in order to isolate more CAFs by outgrowth. Cells were cultured in DMEM or F12 medium (Gibco) supplemented with 10% FBS (Fetal Bovine Serum, Gibco), 2 mmol/L glutamine (Invitrogen), 0.5% sodium pyruvate (Invitrogen), and 1% antibiotic-antimycotic (Invitrogen). The primary CAFs were characterized by immunofluorescence using a positive α -SMA and FAP staining and a negative EpCAM, CD45, KRT19 staining. The first to sixth passages of the primary fibroblasts were used in our experiments. At each passage, β -galactosidase staining (Cell Signaling) was performed according to the manufacturer's recommendations, to ensure cells were not in senescence.

- [1] Leca, J. et al. Cancer-associated fibroblast-derived annexin A6⁺ extracellular vesicles support pancreatic cancer aggressiveness. *J. Clin. Invest.* **126**, 4140-4156 (2016).
- [2] Su, S. et al. CD10⁺GPR77⁺ Cancer-Associated Fibroblasts Promote Cancer Formation and Chemoresistance by Sustaining Cancer Stemness. *Cell* **172**, 841-856 (2018).

Reviewer #3 (Remarks to the Author):

Comment: The authors have addressed my comments carefully by performing many additional experiments.

Response: Thanks to the reviewer's comment.